# I$^2$AM: Interpreting Image-to-Image Latent Diffusion Models via Bi-Attribution Maps

**Junseo Park and Hyeryung Jang** [*]
Department of Computer Science & Artificial Intelligence, Dongguk University

## Abstract

Large-scale diffusion models have made significant advances in image generation, particularly through cross-attention mechanisms. While cross-attention has been well-studied in text-to-image tasks, their interpretability in image-to-image (I2I) diffusion models remains underexplored. This paper introduces Image-to-Image Attribution Maps (**I$^2$AM**), a method that enhances the interpretability of I2I models by visualizing bidirectional attribution maps, from the reference image to the generated image and vice versa. I$^2$AM aggregates cross-attention scores across time steps, attention heads, and layers, offering insights into how critical features are transferred between images. We demonstrate the effectiveness of I$^2$AM across object detection, inpainting, and super-resolution tasks. Our results demonstrate that I$^2$AM successfully identifies key regions responsible for generating the output, even in complex scenes. Additionally, we introduce the Inpainting Mask Attention Consistency Score (IMACS) as a novel evaluation metric to assess the alignment between attribution maps and inpainting masks, which correlates strongly with existing performance metrics. Through extensive experiments, we show that I$^2$AM enables model debugging and refinement, providing practical tools for improving I2I model's performance and interpretability.

## 1 Introduction

Latent diffusion models (LDMs) have recently gained popularity as powerful methods for generating images from random noise with textual description (text-to-image, T2I) (Ramesh et al., 2022; Saharia et al., 2022b; Esser et al., 2024), or image (image-to-image, I2I) (Song et al., 2024; Morelli et al., 2023; Koley et al., 2024). Despite their prevalent adoption, these models have often been developed without a comprehensive investigation into their reliability and interpretability. As these methods are increasingly applied in various fields, ensuring their trustworthy and understandable operation becomes critical. Explainable Artificial Intelligence (XAI) plays a crucial role in addressing this need by providing insights into how and why AI models produce their outputs. For LDMs, XAI methods offer the opportunity to interpret complex internal processes, enabling researchers to identify and mitigate potential issues, enhance model performance, and build user trust. Recent XAI efforts Hertz et al. (2022); Tang et al. (2022); Tumanyan et al. (2023) have largely focused on T2I models, where text inputs are segmented into tokens, allowing for straightforward analysis via cross-attention maps, i.e., how individual tokens influence different parts of the generated output. However, applying similar interpretability methods to I2I models, while promising, presents significant challenges. The spatial and contextual continuity between the input (which we call *reference*) image and the output (which we call *generated*) image complicates this token-wise approach in I2I models. Unlike T2I models, where tokens have discrete and non-spatial relationships, I2I models must account for the complex correlations within and between the reference and generated images. This continuity reveals a considerable gap in our understanding and ability to interpret I2I models from both perspectives: from reference to generated and vice versa.

Despite the challenges of applying token-based methods to I2I models, the shared image domain between the reference and generated images opens new possibilities for bi-directional attribution mapping. Such an approach provides deeper insights into how I2I models capture and transfer visual information between input and output domains, offering a more comprehensive understanding

---

[*]Correspondence to: Hyeryung Jang

of the model's internal processes. In this paper, we address two key research questions that guide our approach to bi-directional attribution. The first question (**Q1**) asks, *"Which regions of the generated image are influenced most by the reference image?"* This focuses on the generated image's perspective, exploring how the model utilizes the reference image during generation. The second question (**Q2**) asks, *"Which parts of the reference image contribute most to the generated image?"* This shifts the focus to the reference image, assessing whether the model captures and effectively transfers critical specifics from the input for generating the output. To answer these questions, we introduce the Image-to-Image Attribution maps (**I²AM**) method. I²AM consolidates cross-attention maps across various axes - such as time steps, attention heads, and layers - to provide a detailed understanding of how the LDM operates. This approach allows us to visualize attribution maps from two perspectives: *(i)* from the reference to the generated image; and *(ii)* from the generated image back to the reference. The distinction between these two maps is crucial: while the former focuses on how the reference image affects specific parts of the generated image, the latter examines how the generated image relates back to the reference. This dual approach enhances our ability to interpret the intricate behavior of I2I diffusion models in various tasks, highlighting the key contributions from each perspective.

To validate the effectiveness of I²AM method, we conducted extensive experiments across various tasks and models, including object detection, inpainting, and super-resolution (Cheng et al., 2022; Yang et al., 2023a;b). Our results demonstrate that I²AM successfully captures critical attribution patterns in each task, offering valuable insights into the underlying generation process. Fig. 1 provides a clear example of an inpainting task, specifically in a virtual try-on, illustrating both directions of influence. The top attribution map shows how different areas of the generated image are influenced by the reference image (**Q1**).

Conversely, the bottom map demonstrates how different regions of the reference image contribute to the generation process (**Q2**). While the bottom map seems to indicate that key information (i.e., logo) is not extracted, the patch-level analysis in the right map illustrates which reference information was extracted during the generation of that grid cell. We also introduced a new evaluation metric for reference-based image inpainting tasks, measuring the consistency between the two attribution maps. This metric offers a reliable way to assess how well the model captures key details from the reference image, showing strong consistency with downstream performance. Finally, we explored the utility of I²AM for debugging and improving I2I models. By examining bi-directional attribution maps, we identified instances where the model failed to capture essential details from the reference or misrepresented certain features in the generated image, adjusting the I2I model in a targeted manner.

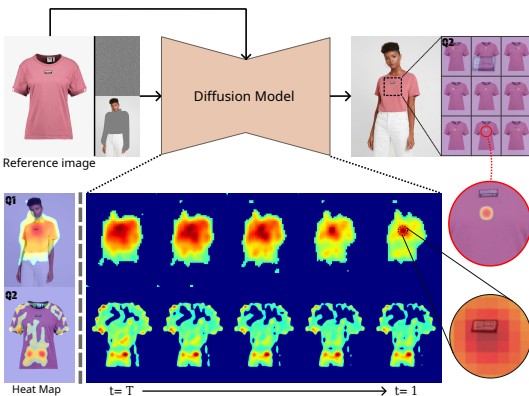

Figure 1: Cross-attention maps using I²AM. The top map shows how the generated image is influenced by the reference image (**Q1**), while the bottom map illustrates how the reference image contributes to the generated image (**Q2**). The right map highlights specific reference-to-output patch contributions.

## 2 RELATED WORK

**Perturbation-based attribution methods.** Perturbation-based methods explain model predictions by altering input features and observing changes in the output, making them suitable for black-box models. Occlusion Sensitivity Zeiler & Fergus (2014b) occludes parts of the input to evaluate their influence on predictions. LIME Ribeiro et al. (2016) perturbs inputs and trains an interpretable local model to approximate the model's predictions, while RISE Petsiuk (2018) generates pixelwise saliency maps using randomized input masking. Although these methods are computationally intensive, they are valuable for analyzing models without requiring access to internal parameters.

**Gradient-based attribution methods.** Gradient-based attribution methods determine input feature importance by analyzing the gradients of the model's output. Early work (Zeiler & Fergus, 2014a; Simonyan, 2013) highlighted the significance of individual input pixels in image-based models.

Class activation maps (Zhou et al., 2016; Selvaraju et al., 2016) further improved visualization by combining gradient and activation maps. Techniques such as SmoothGrad Smilkov et al. (2017) and FullGrad Srinivas & Fleuret (2019) enhanced these visualizations through gradient smoothing and dual importance scoring, while Score-CAM Wang et al. (2020) emphasized global feature contributions. Despite their utility, these methods face challenges like gradient noise and handling negative contributions, which can obscure accurate interpretation.

**Attention-based attribution methods.** With the rise of transformers, *attention-based attribution maps* have become a key focus for understanding model behavior, particularly in text-to-image tasks. These methods intuitively capture the relative importance of tokens, leading to practical applications in areas like image editing (Hertz et al., 2022; Epstein et al., 2023), where object attributes (e.g., style, location, shape) are adjusted by manipulating attention maps. Layout guidance methods (Cao et al., 2023; Kim et al., 2023b; Tumanyan et al., 2023) use structural information encoded in attention maps to refine image generation layouts, while other approaches (Shi et al., 2024; Tewel et al., 2024) ensure subject consistency across multiple images by sharing or transferring attention maps. Semantic correspondence techniques (Zhang et al., 2024; Hedlin et al., 2024) further align cross-attention maps with image content. DAAM Tang et al. (2022) has been instrumental in interpreting cross-attention in text-to-image (T2I) models, revealing the insights of text-conditioned interactions. However, these approaches primarily focus on single-directional token attribution, limiting their applicability to image-to-image (I2I) tasks.

## 3 PRELIMINARIES

**Diffusion models.** Diffusion models Sohl-Dickstein et al. (2015); Ho et al. (2020); Song et al. (2022) are probabilistic generative models that progressively denoise a Gaussian noise sample $\epsilon \sim \mathcal{N}(0, 1)$ to generate data. Starting with a sample $\mathbf{x}_0$ from an unknown distribution $q(\mathbf{x}_0)$, the goal is to train a parametric model $p_{\boldsymbol{\theta}}(\mathbf{x}_0)$ to approximate $q(\mathbf{x}_0)$. These models can be viewed as a sequence of equally weighted denoising autoencoders $\epsilon_{\boldsymbol{\theta}}(\mathbf{x}_t, t)$, which operate over a series of time steps $t = 1 \dots T$. Each step involves predicting a cleaner version of the noisy input $\mathbf{x}_t$, derived from the original input $\mathbf{x}_0$. The training objective is expressed as:

$$\mathcal{L}_{\text{DM}}(\boldsymbol{\theta}) = \mathbb{E}_{\mathbf{x}, \boldsymbol{\epsilon}, t}\left[\|\boldsymbol{\epsilon} - \boldsymbol{\epsilon}_{\boldsymbol{\theta}}(\mathbf{x}_t, t)\|_2^2\right], \tag{1}$$

Latent Diffusion Models (LDMs) Rombach et al. (2022) function similarly but operate in a compressed latent space rather than in the data space. The encoder $\mathcal{E}$ maps the input $\mathbf{x}$ to a latent code $\mathbf{z} = \mathcal{E}(\mathbf{x})$, where noise is added in the latent space to produce $\mathbf{z}_t := \alpha_t \mathcal{E}(\mathbf{x}) + \sigma_t \boldsymbol{\epsilon}$. In image-to-image tasks, the LDM model $\epsilon_{\boldsymbol{\theta}}(\mathbf{z}_t, t, \mathbf{c_I})$ is trained to denoise $\mathbf{z}_t$ using a modified objective:

$$\mathcal{L}_{\text{LDM}}(\boldsymbol{\theta}, \boldsymbol{\phi}) = \mathbb{E}_{\mathbf{z}, \mathbf{c_I}, \boldsymbol{\epsilon}, t}\left[\|\boldsymbol{\epsilon} - \boldsymbol{\epsilon}_{\boldsymbol{\theta}}(\mathbf{z}_t, t, \mathbf{c_I})\|_2^2\right], \tag{2}$$

where $\mathbf{c_I} = \Gamma_{\boldsymbol{\phi}}(\mathbf{I})$ is a conditioning vector derived from a so-called *reference* image $\mathbf{I}$, i.e., input image, via the image encoder $\Gamma_{\boldsymbol{\phi}}$. During training, both $\epsilon_{\boldsymbol{\theta}}$ and $\Gamma_{\boldsymbol{\phi}}$ are jointly optimized to minimize the LDM loss in (2), yielding a reverse process that gradually denoises the noise $\boldsymbol{\epsilon}$ to generate the final output $\mathbf{x}$, referred to as the *generated* image.

## 4 METHODOLOGY: $\text{I}^2\text{AM}$

In this work, we introduce **Image-to-Image Attribution Maps**, which we call $\mathbf{I}^2\mathbf{AM}$, a method aimed at interpreting latent diffusion models using cross-attention maps. Attribution mapping is a powerful tool for analyzing the relationship between specific parts of an image or text (e.g., patches or tokens) and output features. Unlike text-to-image models, which rely on token-based representations, image-to-image generation allows for bidirectional analysis - visualizing two distinct attribution maps - corresponding to two research questions **Q1** and **Q2**, where embeddings of all patches from both the reference and the generated images are required. This level of detailed attribution is challenging for T2I models due to the abstract nature of text tokens. In I2I LDMs, reference and generated images act as both queries and keys in the cross-attention mechanism, allowing us to map the flow of information between the reference and the generated images.

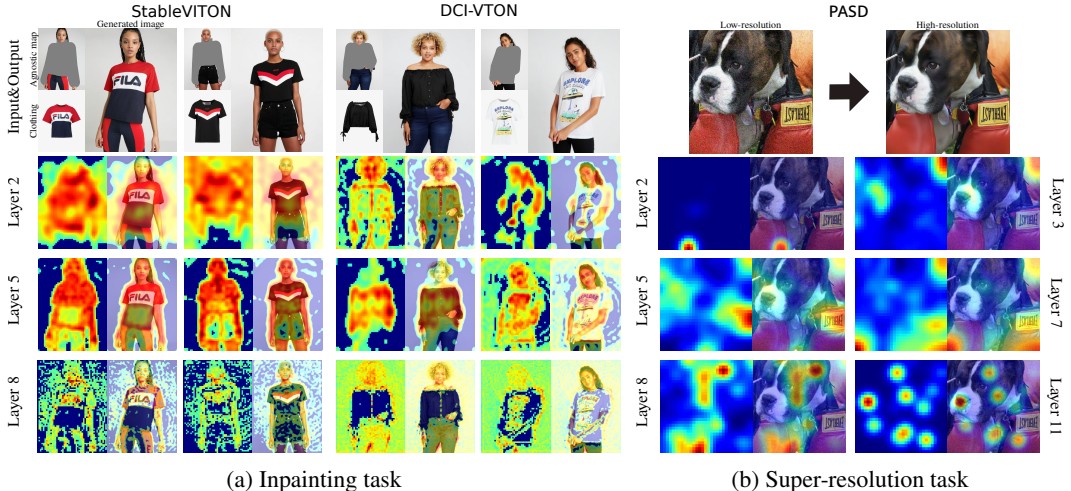

(a) Inpainting task  (b) Super-resolution task

Figure 2: Visualization of layer-level attribution maps (LLAM) for each task. (a) LLAMs for StableVITON and DCI-VTON models at layers 2, 5, and 8 demonstrate how clothing features are progressively incorporated during the inpainting process. (b) LLAMs for PASD model show the contribution of reference data in refining image resolution at different layers.

**Overview of $I^2AM$.** The $I^2AM$ method enables the visualization of bidirectional attribution maps by dividing reference and generated images into smaller patches and analyzing their interactions across three key axes: diffusion time steps, attention heads, and layers. This approach can be applied to various tasks such as segmentation, style transfer, colorization, and depth estimation. By examining each component - time, head, and layer - individually, we gain deeper insights into the model's behavior and its task-specific details. For example, Fig. 2 demonstrates how layer-level analysis aggregates information across time steps and attention heads, revealing broader patterns and attention flows across layers for different tasks. In Fig. 2a, attention scores gradually transit from coarse to fine-grained features as layers deepen, starting with basic color information and even-

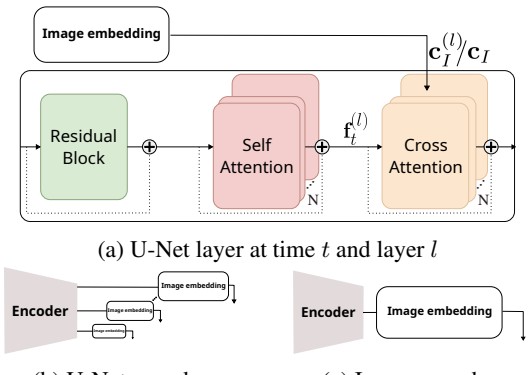

(a) U-Net layer at time $t$ and layer $l$

(b) U-Net encoder  (c) Image encoder

Figure 3: (a) U-Net layer at time $t$ and layer $l$, where the image embeddings are supplied to the cross-attention. (b) U-Net encoder providing multi-scale image embeddings $\mathbf{c}_I^{(l)}$; and (c) image encoder supplying fixed-size embeddings $\mathbf{c}_I$ to the cross-attention module.

tually capturing intricate details like logos and buttons. Conversely, in Fig. 2b, since LQ (low-quality) input signals are provided with the initial noise, the early layers seem to interpret the input as already containing basic information, leading them to pay less attention to coarse features. Further details are provided in the Appendix A.7. This analysis applies to generated and reference images, where maps for generated images are denoted by the subscript g, and those for reference images by r.

## 4.1 BASIC FUNCTIONS

In this subsection, we describe the core operations involved in our $I^2AM$ method.

**Attention score.** At the core of our approach is the computation of attention score, which quantifies the interaction between different regions of the reference and generated images. The attention score is computed as follows:

$$\text{Attn\_Score}(\boldsymbol{Q}, \boldsymbol{K}) = \text{softmax}\left(\frac{\boldsymbol{Q}\boldsymbol{K}^\top}{\sqrt{d_k}}\right), \tag{3}$$

where $\boldsymbol{Q}$ is the query matrix, $\boldsymbol{K}$ is the key matrix, and $d_k$ is the dimension of the key vectors. In our model, both the query and key matrices operate across several axes: time steps $t$, attention heads $n$,

and layers $l$. To capture how the model processes information at different levels within the cross-attention module, we define the attribution map at a specific time step $t$, attention head $n$, and layer $l$ as $\boldsymbol{m}_{t,n}^{(l)}$, which encapsulates the relationships between the reference and generated images.

**Summation operation.** To aggregate information across these axes, we introduce a summation operation allowing aggregation over any subset of the axes $\{t, n, l\}$. The summation is defined as:

$$\text{Sum}_{\mathcal{A}}(\boldsymbol{m}_{t,n}^{(l)}) = \sum_{\mathcal{A}} \boldsymbol{m}_{t,n}^{(l)}, \tag{4}$$

where $\mathcal{A} \subseteq \{t, n, l\}$ represents the axes over which the summation is performed. This operation enables us to analyze the attention behavior across time steps, heads, and layers, providing a comprehensive view of how different patches of reference or generated images contribute to one another.

**Threshold-based transformation.** After the summation, we apply a threshold-based transformation to refine the attention maps. The attention map $\boldsymbol{m}$ is normalized to the range $[0, 1]$, and a threshold $\delta$ is applied to filter out lower values. The final attention map $\boldsymbol{m}$ is computed as:

$$\boldsymbol{m} \leftarrow \boldsymbol{m} \odot \mathbb{I}(\boldsymbol{m} > \delta), \tag{5}$$

where $\mathbb{I}(\cdot)$ is an indicator function that retains values greater than the threshold. This operation ensures that the most relevant attention regions are highlighted while filtering out insignificant areas. The threshold $\delta$ is typically set to $0.4$ to balance visibility and relevance.

## 4.2 UNIFIED-LEVEL ATTRIBUTION MAPS

**Bidirectional attention scores.** In our approach, bidirectional attention scores across time steps, attention heads, and layers form the basis of our operations. As shown in Fig. 3a, the LDM with an $L$-layered cross-attention module derives the pre-cross-attention vectors $\{\mathbf{f}_t^{(l)}\}_{l=1}^L$ at each time step $t$ and layer $l$. Image embeddings of the reference image $\mathbf{I}$ are obtained from various image encoder $\Gamma_\phi$, which can be multi-scale feature vectors $\mathbf{c}_\mathbf{I}^{(l)}$ from U-Net encoders (as in Fig. 3b) or fixed-size embeddings $\mathbf{c}_\mathbf{I}$ from CLIP or DINOv2 (as in Fig. 3c). For simplicity, we use notations of fixed-size embedding $\mathbf{c}_\mathbf{I}$. We adopt a multi-head cross-attention mechanism with $n = 1, \ldots, N$ heads. *Bidirectional* attention scores quantify the interactions between the reference and generated images in two directions. The vectors $\{\mathbf{f}_t^{(l)}\}$ and $\mathbf{c}_\mathbf{I}$ are conditioned to each other through a multi-head cross-attention mechanism.

- **Reference-to-Generated (R2G)** attention score $\mathbf{M}_{\mathsf{g},t,n}^{(l)}$: This score captures the influence of the query $\boldsymbol{Q}$ (reference patch $\mathbf{c}_\mathbf{I}$) on the key $\boldsymbol{K}$ (generated patch $\mathbf{f}_t^{(l)}$) in the key vectors (generated image). This corresponds to **Q1**, measuring how each reference patch influences the generated image.

- **Generated-to-Reference (G2R)** attention score $\mathbf{M}_{\mathsf{r},t,n}^{(l)}$: This score shows how the query (generated patch) corresponds to the key (reference patch) in the key vectors (reference image). This corresponds to **Q2**, measuring how the generated image relates back to the reference image.

The attention scores are computed as:

$$\mathbf{M}_{\mathsf{g},t,n}^{(l)} = \text{Attn\_Score}(\mathbf{W}_{k,n}^{(l)} \mathbf{c}_\mathbf{I}, \mathbf{W}_{q,n}^{(l)} \mathbf{f}_t^{(l)}) \quad \text{and} \quad \mathbf{M}_{\mathsf{r},t,n}^{(l)} = \text{Attn\_Score}(\mathbf{W}_{q,n}^{(l)} \mathbf{f}_t^{(l)}, \mathbf{W}_{k,n}^{(l)} \mathbf{c}_\mathbf{I}), \tag{6}$$

where $\mathbf{W}_{q,n}^{(l)}$ and $\mathbf{W}_{k,n}^{(l)}$ are projection matrices for queries and keys, respectively. While the size of the attention scores may vary across layers, for consistency in aggregation, we resize all attention scores to a common size of $(HW, HW)$, where $H$ and $W$ denote the height and width of the original image $\mathbf{x}_0$. These bidirectional attention maps allow us to analyze the transfer of information between the reference and generated images from both perspectives.

**Unified-level attribution map (ULAM).** To obtain a holistic representation of the attention patterns, we introduce the *unified-level* attribution map (ULAM), which aggregates the attention scores in (6) across $\{t, n, l\}$. Before applying the summation, we first perform a column-wise averaging of the attention scores $\mathbf{M}_{\mathsf{g},t,n}^{(l)}$ and $\mathbf{M}_{\mathsf{r},t,n}^{(l)}$, reducing their dimensions from $(HW, HW)$ to $(HW)$. This

step ensures that the attention maps are more compact and interpretable while retaining the key relationships. The ULAM for R2G and G2R, denoted as $\mathbf{M}_{\mathsf{g}}$ and $\mathbf{M}_{\mathsf{r}}$, are computed as:

$$\mathbf{M}_{\mathsf{g}} = \text{Sum}_{\{t,n,l\}}(\hat{\mathbf{M}}_{\mathsf{g},t,n}^{(l)}), \quad \mathbf{M}_{\mathsf{r}} = \text{Sum}_{\{t,n,l\}}(\hat{\mathbf{M}}_{\mathsf{r},t,n}^{(l)}), \tag{7}$$

where $\hat{\mathbf{M}}_{\mathsf{g},t,n}^{(l)}, \hat{\mathbf{M}}_{\mathsf{r},t,n}^{(l)}$ are the column-wise averaged scores, and the summation is applied over all time steps, heads, and layers. This aggregated ULAM captures the overall contribution of different regions of the reference image to the generated image, and vice versa, measuring relevance loss by examining the alignment of concentrated attention score distributions. After summation, these maps are reshaped to the spatial dimensions $(H, W)$ for visualization, and a threshold-based transformation (5) is applied to filter out lower values, ensuring that the final maps focus on the most critical attention regions.

### 4.3 LAYER/HEAD/TIME-LEVEL ATTRIBUTION MAPS

To fully understand the behavior of diffusion models, it is important to break down the model's core components - time steps, attention heads, and layers. To this end, we introduce three types of attribution maps that offer insights into the role of each of these components in the image generation process. For detailed formulas, please see Appendix A.3.

**Time-level attribution map (TLAM).** The *time-level* attribution map (TLAM) visualizes the generation process over time. The TLAM for a given time group $\tau$ for R2G and G2R directions, denoted as $\mathbf{M}_{\mathsf{g},\tau}$ and $\mathbf{M}_{\mathsf{r},\tau}$, is obtained by summing $\hat{\mathbf{M}}_{\mathsf{g},t,n}^{(l)}$ and $\hat{\mathbf{M}}_{\mathsf{r},t,n}^{(l)}$ over heads and layers for a specific time window $[\tau\Delta t, (\tau+1)\Delta t]$. This results in $T_{\text{group}} = T/\Delta t$ time-level maps. Leveraging the final TLAMs, being applied reshaping and threshold-based transformation, $\{\mathbf{M}_{\mathsf{g},\tau}, \mathbf{M}_{\mathsf{r},\tau}\}_{\tau=0}^{T_{\text{group}}-1}$, we can visualize the gradual formation of the generated image over time.

**Head-level attribution map (HLAM).** The *head-level* attribution map (HLAM) reveals the contributions of different attention heads. For R2G and G2R directions, the HLAM is computed by summing $\hat{\mathbf{M}}_{\mathsf{g},t,n}^{(l)}$ and $\hat{\mathbf{M}}_{\mathsf{r},t,n}^{(l)}$ across time steps and layers, denoted as $\{\mathbf{M}_{\mathsf{g},n}, \mathbf{M}_{\mathsf{r},n}\}_{n=1}^{N}$. After summation and reshaping, we apply threshold-based transformation to obtain the final HLAM. Each head captures different information, enhancing contextual understanding. The model generates semantically aligned outcomes, emphasizing this effect remains unaffected even when integrating the time and layer axes.

**Layer-level attribution map (LLAM).** The *layer-level* attribution map (LLAM) highlights how each layer processes and transforms the input features as they propagate through the network. The LLAMs for R2G direction, denoted as $\{\mathbf{M}_{\mathsf{g}}^{(l)}\}_{l=1}^{L}$, with $L$ being the number of layers, are obtained via summation of $\hat{\mathbf{M}}_{\mathsf{g},t,n}^{(l)}$ and $\hat{\mathbf{M}}_{\mathsf{r},t,n}^{(l)}$ over time steps and heads. The layer-level maps offer a better indicator of model performance across different layers and help uncover areas for potential improvement. On the other hand, analyzing the LLAMs for G2R direction $\mathbf{M}_{\mathsf{r}}^{(l)}$ often proves less informative. This is because it tends to distribute attention more uniformly across the reference image in most layers, mixing irrelevant components (e.g., background and unmasked regions) with relevant information.

**Specific-reference attribution map (SRAM).** Given the challenges associated with interpreting the G2R layer-level attribution map, we propose a more targeted solution: the *specific-reference* attribution map (SRAM). SRAM shows the regions of the reference image that contribute to the generated patch. While the layer-level map $\mathbf{M}_{\mathsf{r}}^{(l)}$ captures the broad interactions from the column-wise averaged scores, SRAM refines this by selecting the relevant $i$-th row from the fundamental attention score $\mathbf{M}_{\mathsf{r},t,n}^{(l)}$ in (6) that corresponds to a specific patch in the generated image. By summing across time steps and attention heads, we construct the SRAM $\{\mathbf{M}_{\mathsf{sr},i}^{(l)}\}_{l=1}^{L}$, which highlights

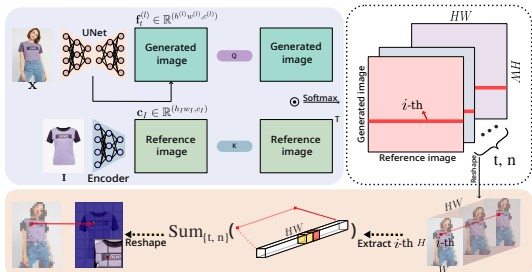

Figure 4: Overview of SRAM, showing how attention scores from all patch embeddings of the reference image (clothing) are calculated to analyze correlation with a specific generated patch $i$. The red point on the clothing indicates the reference patch with the highest influence on the generated image.

how each layer of the model reflects information from the reference image during the generation of a particular ($i$-th) patch in the output image. See Fig. 1 and 4 for illustrative examples.

SRAM provides more concrete positional mapping details between the generated and reference images, making it a valuable feedback signal for model training. For instance, in object inpainting tasks, SRAM guides the model in focusing on the relevant object parts without considering irrelevant background information from the reference image. In super-resolution tasks, it helps improve the sharpness and structural consistency of the generated image by leveraging spatial correspondences between the reference and generated images. Leveraging insights from SRAM can guide the model's learning process by enhancing the roles of specific layers. For example, in layers responsible for capturing important features, training can be improved by strengthening attention mechanisms on patches that are crucial for accurate generation. This focus on critical layers and patches can be used to introduce useful priors into the model, as further explored in Sec. 5.4.

### 4.4 INPAINTING MASK ATTENTION CONSISTENCY SCORE

We introduce the *Inpainting Mask Attention Consistency Score* (IMACS), a metric for evaluating reference-based inpainting tasks in image-to-image latent diffusion models. IMACS measures the alignment between attention maps of generated and reference images ($\mathbf{M_g}$ and $\mathbf{M_r}$) with their respective masks ($\mathbf{x_g}$ and $\mathbf{x_r}$). For example, in a virtual try-on test, the inpainting masks for generated or reference images represent, for example, a clothing-agnostic mask and a cloth mask, respectively. This metric quantifies the consistency between the bidirectional attention maps and the respective inpainting/reference masks, evaluating the effectiveness of information transfer between two images. The score for the R2G attention map, denoted as $\text{IMACS}_g$, is computed as:

$$\text{IMACS}_g = \frac{\sum_{H,W}(\mathbf{M_g} \odot \mathbf{x_g})}{\sum_{H,W}\mathbf{x_g}} - \lambda\frac{\sum_{H,W}(\mathbf{M_g} \odot (\mathbf{1} - \mathbf{x_g}))}{\sum_{H,W}(\mathbf{1} - \mathbf{x_g})}, \tag{8}$$

where $\lambda$ is a penalty factor (default 3), penalizing misaligned attention. The score for the G2R attention map, denoted as $\text{IMACS}_r$, is defined similarly. The higher values of $\text{IMACS}_{g/r}$ indicate better alignment between the attention maps and the corresponding masks, and thus, superior performance as an XAI metric. Increasing $\lambda$ imposes stronger penalties on attention deviating from mask regions, helping to detect issues such as overfocusing on irrelevant areas or neglecting critical regions.

## 5 EXPERIMENTAL RESULTS

We conducted extensive experiments using I$^2$AM across multiple tasks, including object detection, inpainting, and super-resolution, to evaluate its effectiveness in interpreting image-to-image LDMs. These experiments demonstrate the ability of I$^2$AM to enhance interpretability and reveal underlying model behavior across tasks. Experiment details are in Appendix A.4 and A.5. The ablation study of $\delta$ and $\lambda$ for clear map visualization and IMACS consistency is provided in Appendix A.6.

### 5.1 OBJECT DETECTION

In this experiment, we evaluate I$^2$AM's capability in object detection using images generated by the Paint-by-Example (PBE) model on the COCO Lin et al. (2014) dataset. The goal is to assess how effectively I$^2$AM captures and visualizes critical object features in both reference and generated images, even in unseen scenarios. We utilize the unified-level R2G attribution map $\mathbf{M_g}$ to analyze how I$^2$AM retrieves key object regions responsible for object formation and object boundaries obtained by PBE, particularly in complex scenes with multiple objects (Fig. 5).

The comparison with baseline object detection models is shown in Tab. 1. The metric $\text{mIoU}_{\text{GT}}^{>0.5}$ computes the mean Intersection over Union for COCO images where the baseline category score exceeds $0.5$, evaluating the baseline's object detection ability on real images. For PBE-generated images, we use $\text{mIoU}_{\text{gen}}^{>0.5}$ to assess detection performance. While there is a performance drop for generated images compared to ground truth, the PBE-generated images still capture key object features from the reference image, and furthermore, the baselines such as YOLOv3, trained on COCO data, successfully reveal regions critical for object formation. To assess performance in unseen scenarios, we use three baselines: DAAM Tang et al. (2022), 'overall', and 'random$_{10\sim30\%}$'. DAAM

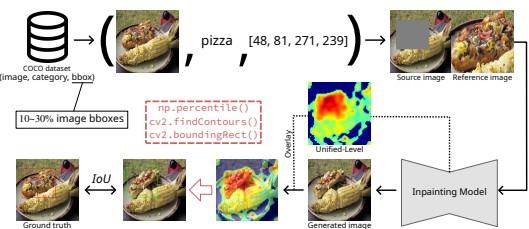

Figure 5: Object detection pipeline using PBE and I²AM. ULAM for R2G direction highlights key regions for object detection in the generated image.

Table 1: Performance comparison of object detection methods, with significant results highlighted in yellow .

| Method | mIoU$_{GT}^{>0.5}$ | mIoU$_{gen}^{>0.5}$ |
|---|---|---|
| Supervised manner & Seen dataset | | |
| Faster-RCNN Ren et al. (2016) | 0.3225 | 0.2658 |
| Mask-RCNN He et al. (2017) | 0.3294 | 0.2706 |
| YOLOv3 Redmon (2018) | 0.2448 | 0.1978 |
| MaskFormer Cheng et al. (2022) | **0.3568** | **0.2932** |
| RTMDet Lyu et al. (2022) | 0.3228 | 0.2516 |
| Unsupervised manner & Unseen dataset | | |
| DAAM Tang et al. (2022) | | 0.1807 |
| Overall | | 0.1807 |
| Random$_{10\sim30\%}$ | | 0.2028 |
| Ours | − | 0.2416 |

is an attention-based T2I method that applies softmax differently from I²AM, producing 'overall' results in PBE. See the Appendix A.7 for details. The 'overall' assumes bounding boxes cover the entire image, while the 'random$_{10\sim30\%}$' compares results with randomly generated bounding boxes covering $10 \sim 30\%$ of the image. Although COCO is a seen dataset for existing baseline methods and unseen for PBE and our method, I²AM achieves a competitive mIOU of $0.2416$ and even outperforms YOLOv3 in certain cases. This highlights I²AM's ability to extract relevant knowledge from reference images, even in challenging environments with generated content.

## 5.2 IMAGE INPAINTING

We evaluate I²AM's interpretability in image inpainting tasks using models like PBE Yang et al. (2023a), DCI-VTON Gou et al. (2023), and StableVITON Kim et al. (2023a). Image inpainting involves filling masked regions of an image by referencing another image, making cross-attention maps essential for understanding how the reference image influences the inpainting process.

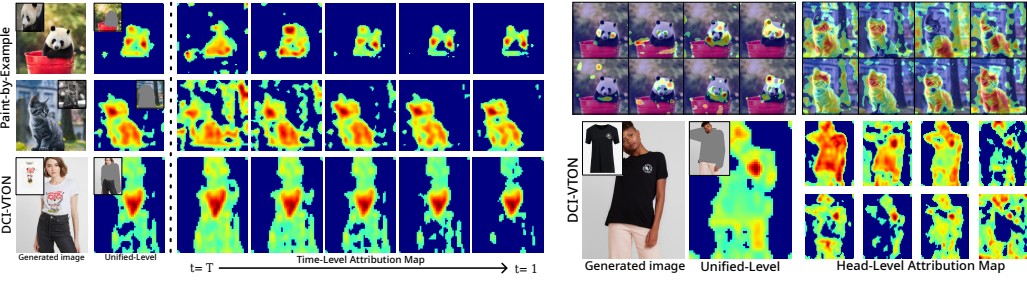

(a) Time-Level Attribution Map $\mathbf{M}_{g,\tau}$

(b) Head-Level Attribution Map $\mathbf{M}_{g,n}$

Figure 6: Attribution maps for R2G. (a) TLAM $\mathbf{M}_{g,\tau}$ visualizes changes in attention patterns over time from $t = T$ to $t = 1$, highlighting where the model focuses at each step. (b) HLAM $\mathbf{M}_{g,n}$ shows the contributions of the 8 attention heads, comparing how each head focuses on specific regions of the image.

To address **Q1**, we first analyze the R2G map, focusing on the areas where the reference image contributes most to the filled-in regions in the generated output. The time-level attribution map (TLAM) $\mathbf{M}_{g,\tau}$, illustrated in Fig. 6a, shows how the model progressively refines object structure over time, assigning higher attention to crucial details like facial features and clothing logos. As the inpainting process unfolds, the model transitions from low-frequency to high-frequency features. The head-level map (HLAM) $\mathbf{M}_{g,n}$, displayed in Fig. 6b, reveals variations across attention heads, with some attention extending beyond the masked area. Integrating information beyond the masked region may result in the loss of certain features but also improves contextual understanding, enabling semantically consistent outputs. Since HLAM was expected to capture diverse information, this behavior is seen as a positive outcome. Fig. 6 demonstrates the capacity of the LDM to recognize and integrate key patterns throughout the inpainting process.

Next, we extend this analysis in Fig. 7 for StableVITON model, using bidirectional attention maps. The time-level G2R maps $\mathbf{M}_{r,\tau}$, shown on the left, indicate minimal changes over time, while the head-level G2R maps $\mathbf{M}_{r,n}$ highlight diversity across heads, with a common focus on logos. In contrast, the R2G maps in the middle panel show how details such as clothing patterns and textures are progressively transferred from the reference to the generated image, as reflected in $\mathbf{M}_{g,\tau}$. Finally,

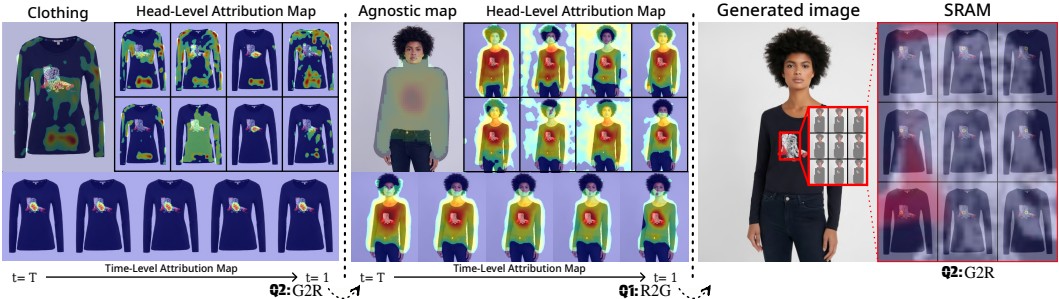

Figure 7: Generation flow of StableVITON through $I^2AM$. The traits derived from G2R maps are shown (left), and the usage is displayed in R2G maps (middle). (right) We visualize $\mathbf{M}_{sr,i}^{(5)}$, capturing key blending features that link basic properties with finer details.

the SRAM $\mathbf{M}_{sr,i}^{(5)}$, shown on the right, provides positional mapping at layer 5, indicating how clothing colors and patterns from the reference image are integrated into the final output. Additional results are provided in the Appendix for further details.

## 5.3 SUPER-RESOLUTION

In this section, we evaluate the PASD Yang et al. (2023b) model for super-resolution tasks, where the goal is to enhance the resolution of a low-resolution image by progressively refining it. Unlike inpainting, which reconstructs specific masked regions, super-resolution aims to improve the overall image resolution while preserving key details from the input. First, Fig. 8 provides insights into both **Q1** and **Q2** using bidirectional (R2G and G2R) maps. The HLAMs $(\mathbf{M}_{g,n}, \mathbf{M}_{r,n})$ reveal that different heads handle distinct regions of text descriptions from the reference image but work together to enhance fine details in the final output. The TLAMs $(\mathbf{M}_{g,\tau}, \mathbf{M}_{r,\tau})$ show that the PASD model consistently focuses on critical image features, e.g., text descriptions, throughout the process, maintaining stable attention across time steps. The consistent attention distribution across time and heads highlights the model's reliance on detailed information from the low-resolution input to improve the high-resolution image.

Fig. 9 extends this analysis to a layer-level view using the SRAMs $(\mathbf{M}_{sr,i}^{(l)})$, illustrating how patches at similar positions across different layers contribute to super-resolution. Layers 2 to 7 show precise attention alignment, indicating the model's effectiveness in preserving critical image features from the low-resolution input at appropriate locations. However, attention scores in higher layers (e.g., layers 9 and 11) display slight misalignments. This suggests that while the lower layers focus on refining key structural features and capturing accurate positional information, the higher layers may struggle with processing more abstract and challenging patterns. As also shown in Fig. 2, each layer performs a distinct function, with some layers prioritizing fine details while others focus on higher-level semantics. We note that similar results are observed in another super-resolution model, SeeSR (see Appendix A.7).

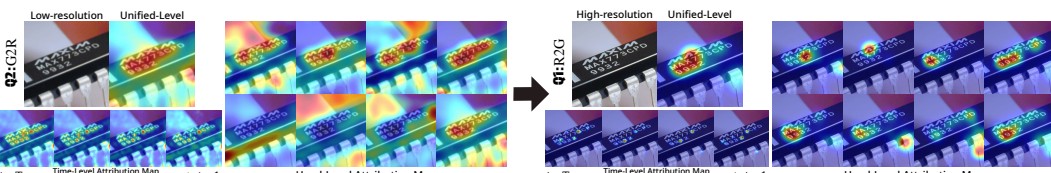

Figure 8: Improvement process of super-resolution, showing how content from the G2R map is meaningfully incorporated into the R2G map. Progressive refinement of details from low to high resolution is visualized through head- and time-level maps.

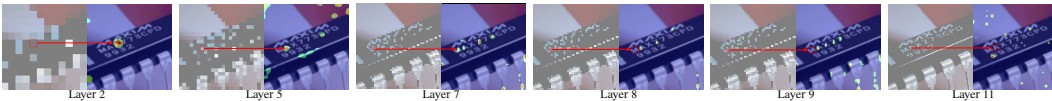

Figure 9: Gradual change in attention within SRAM $\mathbf{M}_{sr,i}^{(l)}$, at corresponding positions in multiple layers. Lower layers achieve intuitive position alignment, while higher layers manage more complex contexts. Zoom in to check for position alignment.

| Method | FID↓ | KID↓ | LPIPS↓ | SSIM↑ | IMACS$_g$↑ | IMACS$_r$↑ |
|---|---|---|---|---|---|---|
| DCI-VTON Gou et al. (2023) | 13.0953 | 0.0334 | 0.0824 | 0.8612 | 0.0785 | – |
| StableVITONKim et al. (2023a) | **10.6755** | 0.0064 | **0.0817** | 0.8634 | **0.3083** | **0.3388** |
| Custom | 11.6572 | 0.0042 | 0.1020 | 0.8396 | 0.0833 | 0.0403 |
| Refined custom | 11.5420 | **0.0022** | 0.0964 | **0.8644** | 0.2215 | 0.0948 |

Table 2: Quantitative comparison on VITON-HD dataset. Evaluation metrics include FID, KID, LPIPS, and SSIM for downstream tasks, alongside our proposed XAI evaluation metric, IMACS (with $\delta = 0.4, \lambda = 3$). **Bold** and underline indicate the best and second-best result, respectively.

### 5.4 APPLICATIONS: IMACS AND MODEL DEBUGGING

This section explores the utility of our proposed metrics IMACS$_{g/r}$ for model debugging and performance improvement. IMACS, alongside traditional metrics like FID, KID, LPIPS, and SSIM, provides deeper insights into attention alignment between reference and generated images.

Our analysis highlights variations in attention dispersion within LDMs, particularly in ULAMs ($\mathbf{M}_g, \mathbf{M}_r$) during inpainting tasks, where mask alignment is critical. For example, DCI-VTON exhibited scattered attention scores, leading to color discrepancies, information loss from the reference, and unnecessary patterns (see Fig. 6), degrading the model's consistency with the reference image. In contrast, StableVITON showed better alignment, producing more accurate and consistent results. This improved alignment was reflected in higher IMACS scores (Tab. 2), which directly correlated with other metrics. StableVITON achieved the highest alignment, followed by the refined custom model, the original custom model, which will be discussed below, and DCI-VTON. These results demonstrate a clear relationship between higher IMACS scores and better task performance, confirming the metric's effectiveness in evaluating model accuracy.

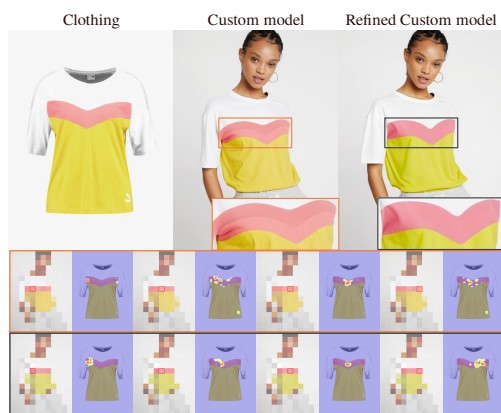

Specific-Reference Attribution Map

We also used IMACS for model debugging. Our experiments confirmed that each layer plays distinct roles in inpainting (Fig. 2). As illustrated in Fig. 7, specific layers in well-trained models like StableVITON effectively map spatial information. In our custom model, detailed in Appendix A.5, layer 2 exhibited similar capabilities but suffered from score dispersion and geometric misalignment SRAM $\mathbf{M}_{sr,i}^{(2)}$. To address this, we applied loss functions aimed at i) densifying attention distributions and ii) improving semantic alignment. As shown in Fig. 10, the refined custom model corrected color inconsistencies and enhanced attention accuracy, resulting in better overall performance, as also observed in Tab. 2.

Figure 10: Comparative analysis of the custom model before and after improvements based on SRAM debugging. The orange box highlights color confusion in the initial model, while the black box shows improved color consistency in the refined model. The SRAM illustrates how debugging improved performance.

### 6 CONCLUSION

We propose the Image-to-Image Attribution Maps (I$^2$AM), a method for bi-directional analysis of how I2I models transfer visual information between input and output domains. By aggregating cross-attention maps across time steps, attention heads, and layers, I$^2$AM generates two attribution maps: one capturing the influence of the reference image on the generated image, and another showing how the generated image relates back to the reference. Our experiments across object detection, inpainting, and super-resolution tasks demonstrate that I$^2$AM effectively enhances model interpretability and captures critical attribution patterns. Moreover, I$^2$AM provides valuable insights for model debugging and refinement. While primarily tested in paired settings with reference-based tasks, preliminary results indicate its potential for unpaired scenarios. Future work aims to extend I$^2$AM to broader applications, such as colorization and style transfer, to further expand its utility.

ACKNOWLEDGMENTS

This research was supported by the MSIT(Ministry of Science and ICT), Korea, under the ITRC(Information Technology Research Center) support program(IITP-2025-2020-0-01789), and the Artificial Intelligence Convergence Innovation Human Resources Development(IITP-2025-RS-2023-00254592) supervised by the IITP(Institute for Information & Communications Technology Planning & Evaluation).

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

# A APPENDIX

## A.1 RELATED WORK

**Image inpainting with text-to-image diffusion models.** Beyond traditional image inpainting techniques that employ noisy backgrounds outside mask regions, text-to-image diffusion models such as GLIDE Nichol et al. (2021) integrate masked images with text prompts. This innovation enhances the diffusion process, using information from outside the mask to improve both contextual relevance and visual consistency. Similarly, Blended diffusion Avrahami et al. (2022) utilizes CLIP scores to align generated images with text prompts, and SmartBrush Xie et al. (2023) is proposed to predict object shapes within a masked area, thus preserving the integrity of the surrounding background.

**Image-to-image diffusion-based image inpainting.** Image-to-image diffusion models for inpainting are less common than text-to-image models. Notable examples include Palette Saharia et al. (2022a) and Paint-by-Example (PBE) Yang et al. (2023a). Palette serves as a general framework for image-to-image translation, yielding accurate results for inpainting but cannot be visualized with our proposed method due to its concatenated image input. To reduce self-referencing in image generation, PBE uses strong augmentation and only the CLS token from the CLIP image encoder Radford et al. (2021) to focus on relevant objects while ignoring background noise. Since PBE provides image embeddings with cross-attention, our method can be applied. Additionally, StableVITON Kim et al. (2023a) and DCI-VTON Gou et al. (2023) excel in the specialized task of virtually dressing clothes and can also utilize our methodology by integrating ControlNet Zhang et al. (2023) structures and warping networks Ge et al. (2021).

## A.2 PRELIMINARIES

**Image inpainting.** This task involves controlling image editing using semantic masks. While traditional image inpainting Lugmayr et al. (2022) focused solely on filling masked areas, recent approaches, like multi-modal image inpainting Xie et al. (2023); Nichol et al. (2021); Avrahami et al. (2022); Couairon et al. (2022); Yu et al. (2023), use guidance such as text or segmentation maps to fill masked regions. The main focus of this paper, VITON, is a type of image inpainting where clothes are virtually worn on a person. The unique aspect is that while maintaining the pose, body shape, and identity of the person, the clothing product must seamlessly deform to the desired clothing area. Additionally, preserving the details of the clothing product is a requirement.

**Classfier-free guidance.** Classifier-free guidance Ho & Salimans (2022) (CFG) is a method for trading off the quality and diversity of samples generated by diffusion models. It is commonly used in text, class, and image-conditioned image generation to enhance the visual quality of generated images and create sampled images that better match the conditions. CFG effectively shifts probability towards data achieving high likelihood for the condition $\mathbf{c}_I$. Training for unconditional denoising involves setting the condition to a null value at regular intervals. At inference time, the guide scale $s$ is set to $s \geq 1$, extrapolating the modified score estimate $\hat{\epsilon}$ towards the conditional output $\epsilon_c$ while moving away from the unconditional output $\epsilon_{uc}$ direction.

$$\hat{\epsilon} = \epsilon_{uc} + s(\epsilon_c - \epsilon_{uc}), \tag{9}$$

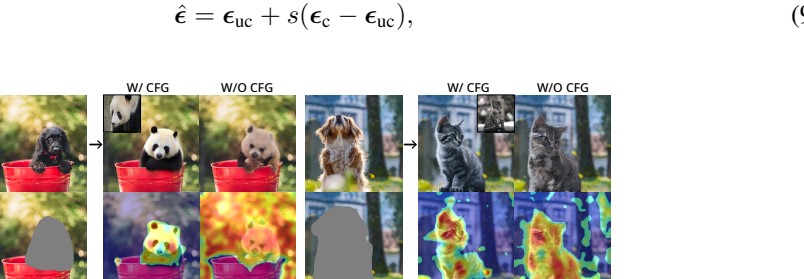

Figure 11: R2G visualization results with and without CFG, using PBE. The dispersion of attention scores exceeded the inpainting mask's range when CFG was not used.

The diffusion model relies on Classifier-free Guidance (CFG) technology, as shown in Fig. 11, depicting the time-and-head integrated attribution map based on the presence of CFG. CFG assigns

a higher likelihood to the reference image, shifting the output accordingly. This shifted output is repeatedly utilized as input for the model, resulting in a shift in the distribution of the attribution map. With the application of CFG, the model better reflects the reference image, facilitating the synthesis of images in appropriate regions.

### A.3 LAYER/HEAD/TIME-LEVEL ATTRIBUTION MAPS

This subsection presents the derivation of the omitted equations and additional samples.

**Time-level attribution map.** TLAM for each group $\{\mathbf{M}_{\mathsf{g},\tau}, \mathbf{M}_{\mathsf{r},\tau}\}_{\tau=0}^{T_{\text{group}}-1} \in \mathbb{R}^{(H,W)}$ provide how the model gradually constructs the image by monitoring its generation process over time.

$$\hat{\mathbf{M}}_{\mathsf{g},t,n}^{(l)} = \frac{1}{HW}\sum_{i=1}^{HW}\mathbf{M}_{\mathsf{g},t,n}^{(l)}[i,:], \quad \hat{\mathbf{M}}_{\mathsf{r},t,n}^{(l)} = \frac{1}{HW}\sum_{i=1}^{HW}\mathbf{M}_{\mathsf{r},t,n}^{(l)}[i,:] \tag{10}$$

$$\mathbf{M}_{\mathsf{g},t} = \text{Sum}_{\{n,l\}}(\hat{\mathbf{M}}_{\mathsf{g},t,n}^{(l)}), \quad \mathbf{M}_{\mathsf{r},t} = \text{Sum}_{\{n,l\}}(\hat{\mathbf{M}}_{\mathsf{r},t,n}^{(l)}) \tag{11}$$

$$\mathbf{M}_{\mathsf{g},\tau} = \sum_{t=\tau\cdot\Delta t}^{(\tau+1)\cdot\Delta t}\mathbf{M}_{\mathsf{g},t}, \quad \mathbf{M}_{\mathsf{r},\tau} = \sum_{t=\tau\cdot\Delta t}^{(\tau+1)\cdot\Delta t}\mathbf{M}_{\mathsf{r},t} \tag{12}$$

**Head-level attribution map.** HLAM $\{\mathbf{M}_{\mathsf{g},n}, \mathbf{M}_{\mathsf{r},n}\}_{n=1}^{N} \in \mathbb{R}^{(H,W)}$ reveals the contributions of each attention head to different regions of the generated image. A varied distribution of heads suggests effective detection and emphasis on multiple features, with the primary object consistently scoring high.

$$\mathbf{M}_{\mathsf{g},n} = \text{Sum}_{\{t,l\}}(\hat{\mathbf{M}}_{\mathsf{g},t,n}^{(l)}), \quad \mathbf{M}_{\mathsf{r},n} = \text{Sum}_{\{t,l\}}(\hat{\mathbf{M}}_{\mathsf{r},t,n}^{(l)}) \tag{13}$$

**Layer-level attribution map.** LLAM $\{\mathbf{M}_{\mathsf{g}}^{(l)}\} \in \mathbb{R}^{(H,W)}$ provides insights into how each layer captures, transforms, and interacts with features, highlighting their impact on the final generated image while revealing potential areas for improvement.

$$\mathbf{M}_{\mathsf{g}}^{(l)} = \text{Sum}_{\{t,n\}}(\hat{\mathbf{M}}_{\mathsf{g},t,n}^{(l)}) \tag{14}$$

**Specific-Reference Attribution Map.** SRAM $\{\mathbf{M}_{\mathsf{sr},i}^{(l)}\}_{l=1}^{L} \in \mathbb{R}^{(H,W)}$ identifies which areas of the reference image a specific patch of the generated image examines, utilizing $\mathbf{M}_{\mathsf{r},t,n}^{(l)}$, represented as a matrix with dimensions $(HW, HW)$.

$$\mathbf{M}_{\mathsf{sr},i}^{(l)} = \text{Sum}_{\{t,n\}}(\mathbf{M}_{\mathsf{r},t,n}^{(l)}[i,:]) \tag{15}$$

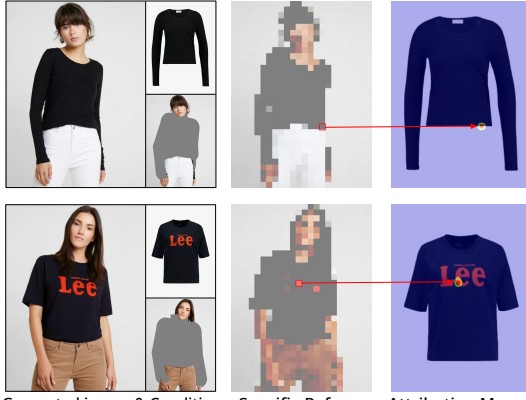

Generated image & Condition   Specific-Reference Attribution Map

Figure 12: Specific-Reference Attribution Map $\mathbf{M}_{\mathsf{sr},i}^{(5)}$ visualization. The information from the reference image that the small red box during synthesis referenced is indicated by the red arrows.

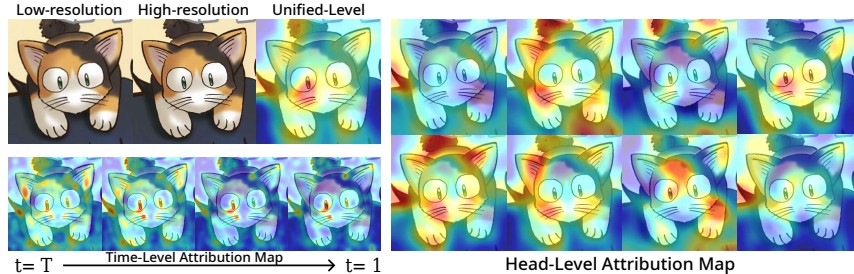

Figure 13: Attribution maps for generated image (R2G) in super-resolution.

## A.4 IMPLEMENTATION DETAILS

**Datasets.** Paint-by-Example (PBE) was trained on the OpenImages Kuznetsova et al. (2020). It consists of 16 million bounding boxes for 600 object classes across 1.9 million images. StableVITON and DCI-VTON were trained on VITON-HD Choi et al. (2021). It is a dataset for high-resolution (i.e., $1024 \times 768$) virtual try-on of clothing items. Specifically, it consists of $13,679$ frontal-view woman and top clothing image pairs are further split into $11,647/2,032$ training/testing pairs.

**Evaluation.** We evaluated DCI-VTON, StableVITON, and a custom model using IMACS. To demonstrate the consistency of IMACS with downstream tasks, we additionally employ evaluation metrics from the VITON task: FID, KID, SSIM, and LPIPS. Specifically, we use paired settings, where person images wearing reference clothing are available, and unpaired settings, where person images wearing reference clothing are not available. In paired settings, we apply SSIM and LPIPS to measure the similarity between the two images, while in unpaired settings, we use FID and KID to measure the statistical similarity between real and generated images. Lower scores for FID, KID, and LPIPS, and higher scores for SSIM and IMACS, indicate better performance. We conduct all evaluations on images of size $512 \times 384$ and all figures are visualized using $I^2AM$ introduced in Section 4.

**Existing models.** We studied the three inpainting models: Paint-by-Example (PBE) Yang et al. (2023a), StableVITON Kim et al. (2023a), and DCI-VTON Gou et al. (2023). While PBE and DCI-VTON follow the architecture in Fig. 3c, StableVITON follows Fig. 3b. Unlike PBE and DCI-VTON, which rely on the CLS token, StableVITON utilizes all patch embeddings from the reference image, allowing for bidirectional visualization. Similarly, the super-resolution models PASD Yang et al. (2023b) and SeeSR Wu et al. (2024) adopt the structure in Fig. 3b and enable bidirectional visualization through cross-attention on all patch embeddings. The number of cross-attention layers ($L$) is determined by the decoder blocks, as interactions between input and reference data are primarily learned during the decoding stage. PBE, StableVITON, DCI-VTON, SeeSR, and our custom model have 9 cross-attention layers, while PASD has 15. Each model used $T = 50$ and $T_{group} = 5$ with DDIM Song et al. (2022) sampler, except for PASD, which had $T = 20$ and $T_{group} = 4$. Most models used 8 attention heads, while SeeSR had varying attention heads: 20 for resolution 16, 10 for resolution 32, and 5 for resolution 64. CFG scales $s$ of $5, 5, 1, 5.5, 5$ and 9, respectively. We visualized the reverse SRAM $M_{sr}^{(l)}$ for layer 5 in StableVITON. Pre-trained models were obtained from their respective GitHub repositories. However, due to limited samples for $IMACS_r$ measurement, demonstrating its consistency with downstream task performance was challenging. This led to the development of a custom model utilizing all patch embeddings from reference images, extending the approach of StableVITON and PASD to other image-to-image latent diffusion models.

## A.5 CUSTOM MODEL

The custom model, based on Stable Diffusion v1.5, replaces the CLIP text encoder with a large image encoder that uses all patch embeddings. It follows the architecture in Fig. 3c and was fine-tuned on the VITON-HD dataset. The model takes a person image $\mathbf{x} \in \mathbb{R}^{(H,W,3)}$, a clothing-agnostic person representation $\mathbf{x}_a \in \mathbb{R}^{(H,W,3)}$, a dense pose $\mathbf{x}_d \in \mathbb{R}^{(H,W,3)}$, and a clothing image $\mathbf{I} \in \mathbb{R}^{(H,W,3)}$, filling the agnostic map $\mathbf{x}_a$ with the clothing image (reference image) $\mathbf{I}$. The input to

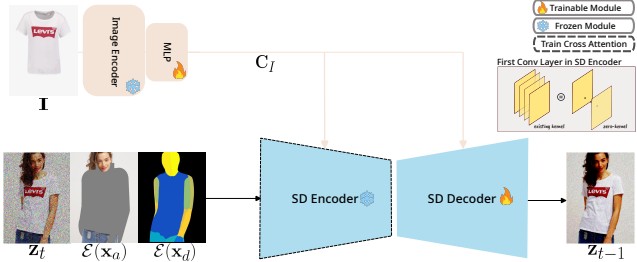

Figure 14: An overview of custom model. For VITON task, the model takes three inputs: the noisy image, agnostic map, and dense pose. Image embeddings serve as the key and value for the cross-attention.

the U-Net expands the initial convolution layer to 12 channels ($4 + 4 + 4 = 12$), initialized with zero weights. All components except $\mathbf{z}$ and $\mathbf{I}$ pass through the encoder $\mathcal{E}$. The custom model overview is shown in Fig. 14. The model was trained for 90 epochs with a batch size of 64. DDIM was used for 50 steps with $T_{\text{group}} = 5$, and a CFG scale ($s$) of 5. The custom model has 8 attention heads ($N = 8$) and 9 cross-attention layers ($L = 9$), with SRAM visualizations focusing on layer 2.

We observed that similar to StableVITON in layer 2, spatial mapping occurs between reference patches and generated patches. However, we identified misalignment in the SRAM's attention scores for each patch and incorporated this into the loss function for training. The goal is to improve color consistency by accurately matching the semantic correspondence of each patch. The proposed loss function consists of three components: $\mathcal{L}_{\text{DCML}}$, $\mathcal{L}_{\text{TV}}$, and $\mathcal{L}_{\text{CWG}}$. Given the standard attention map $A \in \mathbb{R}^{(H', W', h', w')}$ (where the query is the generated image and the key is the reference image), 2D coordinate map $D \in \mathbb{R}^{(h', w', 2)}$, and ground truth warped clothing mask $W \in \{0, 1\}^{(H', W')}$, the loss functions are computed as follows. Note that these loss functions are applied exclusively to the warped clothing region.

- **Weighted Center Coordinate Map**: Calculate the weighted center coordinates $C \in \mathbb{R}^{(H', W', 2)}$ by combining the attention map and coordinate map at each position:

$$C_{ijn} = \frac{1}{h'w'} \sum_{k=1}^{h'} \sum_{l=1}^{w'} (A_{ijkl} \odot D_{kln}), \tag{16}$$

- **Distance-Centering Maximization Loss** ($\mathcal{L}_{\text{DCML}}$): $\mathcal{L}_x$ adjusts the coordinate $x$ by pushing the centers to the right within the same row, while $\mathcal{L}_y$ adjusts $y$ by pushing the centers below within the same column. This aligns the relative positions of patch centers, promoting semantic correspondence and spatial consistency:

$$\mathcal{L}_x = \sum_{i=1}^{H'} \sum_{j=1}^{W'-1} \sum_{k=j+1}^{W'} \max(0, C_{i,j,0} - C_{i,k,0}), \tag{17}$$

$$\mathcal{L}_y = \sum_{j=1}^{W'} \sum_{i=1}^{H'-1} \sum_{k=i+1}^{H'} \max(0, C_{i,j,1} - C_{k,j,1}), \tag{18}$$

$$\mathcal{L}_{\text{DCML}} = \mathcal{L}_x + \mathcal{L}_y, \tag{19}$$

- **Total Variation Loss** ($\mathcal{L}_{\text{TV}}$): Mitigates abrupt changes caused by $\mathcal{L}_{\text{DCML}}$, ensuring smooth transitions and maintaining appropriate spacing between patches:

$$\mathcal{L}_{\text{TV}} = \|\nabla C\|_2^2, \tag{20}$$

- **Center-weighted Gaussian Loss** ($\mathcal{L}_{\text{CWG}}$): Encourage the attention scores for each generated patch to focus on the center coordinates using a Gaussian-based distance calculation.

The standard deviation $\sigma$ is set to 1:

$$\Delta_{i,j,k,l} = \sqrt{(C_{i,j,0} - k)^2 + (C_{i,j,1} - l)^2}, \quad \Delta_{i,j,k,l} \in \mathbb{R}^{(H',W',h',w')}, \quad (21)$$

$$\mathcal{L}_{\text{CWG}} = -\frac{1}{H'W'h'w'} \sum_{i=1}^{H'} \sum_{j=1}^{W'} \sum_{k=1}^{h'} \sum_{l=1}^{w'} \exp\left(-\frac{\Delta_{i,j,k,l}^2}{2\sigma^2}\right) \odot A_{i,j,k,l}, \quad (22)$$

These loss functions help learn the precise correspondence between patches and improve the consistency and quality of the generated image. $\mathcal{L}_{\text{DCML}}$ aligns semantic correspondence between patches, $\mathcal{L}_{\text{TV}}$ enhances spatial consistency, and $\mathcal{L}_{\text{CWG}}$ ensures the attention map focuses on the center coordinates, ultimately yielding more accurate and consistent results. Finally, the overall loss function $\mathcal{L}$ combines these components along with the baseline latent diffusion model loss $\mathcal{L}_{\text{LDM}}$ to optimize all aspects of the model's performance jointly:

$$\mathcal{L} = \mathcal{L}_{\text{LDM}} + \lambda_{\text{DCML}}\mathcal{L}_{\text{DCML}} + \lambda_{\text{TV}}\mathcal{L}_{\text{TV}} + \lambda_{\text{CWG}}\mathcal{L}_{\text{CWG}}, \quad (23)$$

Where $\lambda_{\text{DCML}} = 0.01$, $\lambda_{\text{TV}} = 0.0001$, and $\lambda_{\text{CWG}} = 2$, these values represent the strength of each loss term during training.

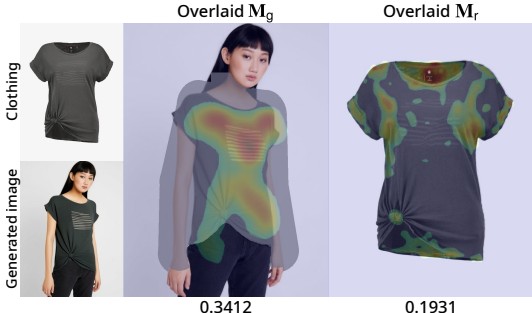

Figure 15: Example of a refined custom model with the $\text{IMACS}_{\text{g/r}}$ metric. Each attribution map shows high attention scores aligned with its mask region. The scores below the figures are $\text{IMACS}_{\text{g/r}}$ values measured with $\lambda$ set to 3.

### A.6 ABLATION STUDY

An ablation study was conducted to find the optimal threshold for filtering small values to improve $\text{I}^2\text{AM}$'s clarity and accuracy. Tab. 3 shows the results of adjusting the threshold values while performing object detection with $\text{I}^2\text{AM}$ on PBE. A threshold $\delta = 0.2$ sets values below 0.2 to zero in a $[0, 1]$-normalized map, with the highest $\text{mIoU}_{\text{gen}}^{>0.5}$ achieved at the values of 0.4 (default) and 0.5. Additionally, we examined the effect of changing the IMACS penalty factor $\lambda$ on the correlation with downstream task metrics, as shown in Tab. 4. A larger $\lambda$ resulted in a greater decrease in IMACS for models with significant information loss from the reference image, as observed in DCI-VTON. Furthermore, we confirmed that the correlation remained stable even with changes in the penalty factor $\lambda$.

### A.7 ADDITIONAL RESULTS

**Comparison with DAAM.**

The primary difference between $\text{I}^2\text{AM}$ and T2I attention map visualization methods like DAAM Tang et al. (2022) and prompt-to-prompt Hertz et al. (2022) lies in the direction of softmax

| Method | $\text{mIoU}_{\text{gen}}^{>0.5}$ | | | | |
|---|---|---|---|---|---|
| | $\delta = 0.2$ | 0.3 | 0.4 | 0.5 | 0.6 |
| Ours | 0.2413 | 0.2413 | **0.2416** | **0.2416** | 0.24 |

Table 3: Ablation study of the threshold $\delta$ in the attention maps. The default $\delta$ is 0.4

| Method | $\lambda = 1$ | | 3 (default) | | 5 | | 10 | |
|---|---|---|---|---|---|---|---|---|
| | IMACS$_g$ ↑ | IMACS$_r$ ↑ | IMACS$_g$ | IMACS$_r$ | IMACS$_g$ | IMACS$_r$ | IMACS$_g$ | IMACS$_r$ |
| DCI-VTON | 0.4291 | – | 0.0785 | – | $-0.3051$ | – | $-1.2155$ | – |
| StableVITON | **0.5376** | **0.3534** | **0.3083** | **0.3388** | **0.07** | **0.3211** | **-0.1219** | **0.301** |
| Custom | 0.2352 | 0.0677 | 0.0833 | 0.0403 | $-0.1901$ | 0.0091 | $-0.4579$ | $-0.0527$ |
| Refined custom | 0.4675 | 0.1249 | 0.2215 | 0.0948 | 0.029 | 0.0513 | $-0.3105$ | 0.0112 |

Table 4: Ablation study of the penalty factor $\lambda$ on the IMACS metric. As $\lambda$ increases, the punishment for attention scores outside the masked region intensifies. A larger $\lambda$ indicates that the model loses more information, as the value drop becomes more significant. **Bold** and underline indicate the best and second-best result, respectively.

application. T2I models allow visualization only of the R2G attribution map, which we compare with $\mathbf{M}_{g,t,n}^{(l)}$ from equation 3. DAAM uses a standard approach to calculate attention scores, treating the generated image as the Query and the prompt as the Key. This approach separates tokens, highlighting those with the most influence on the generated patches, while automatically reducing the values of less important tokens. In contrast, the contextual continuity of the reference image makes it impossible to separate, so we focus on how it influences the generated patches as a whole. We treat the reference image as a single token and reverse the softmax direction, using it as the Query in equation 3. This allows us to understand its influence on the generated patches holistically. However, this approach does not identify which specific reference patches contribute information.

To assess the relevance of the reference information, we visualize the G2R attribution map by treating the generated image as a single token. G2R provides a similar analysis to T2I by comparing reference patches and complements R2G. However, if irrelevant areas dominate the generated image, distortions may arise in G2R. To address this, we propose SRAM $\mathbf{M}_{sr,i}^{(l)}$, which identifies which generated patches extract meaningful information from the reference. SRAM can be computed for all generated patches and aids in interpreting their significance.

Applying softmax in DAAM's direction to I2I models cannot produce interpretable maps. Using DAAM's softmax on the generated image results in uniform attention values across pixels, as the row sums of the softmax are 1 (see Fig. 16). Fig. 16a shows ULAM $\mathbf{M}_g$ computed with StableVITON using all patch embeddings. While attention scores should theoretically be identical, cumulative errors from resizing, normalization, and floating-point inaccuracies introduce discrepancies, resulting in a meaningless map. Fig. 16b shows ULAM $\mathbf{M}_g$ computed with only the CLS embedding, which eliminates these errors and assigns identical scores across all regions. This result matches the 'overall' row in Tab. 1 for object detection tasks. In contrast, by reversing the softmax direction, our approach can visualize the R2G attribution map using only the CLS embedding (e.g., PBE and DCI-VTON).

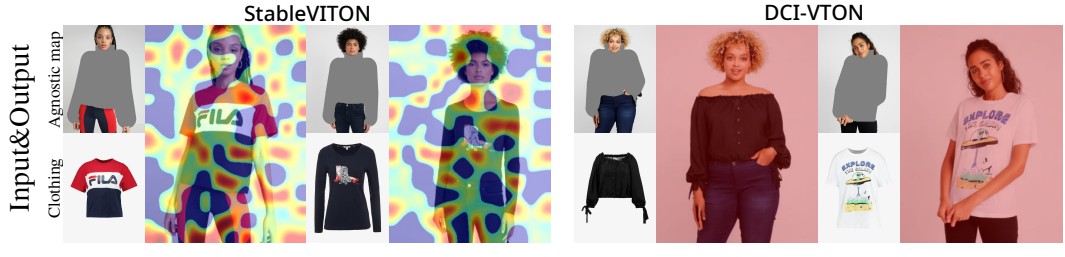

(a) All patch embeddings          (b) Only CLS embedding

Figure 16: Unified-Level Attribution Map $\mathbf{M}_g$ results using DAAM. (a) In the process of aggregating all patch embeddings from the reference image, error accumulates, resulting in a meaningless attention map. (b) Visualization using the reference image's CLS embeddings. In the error-free case, all attention values are the same.

**Comparison of PASD and SeeSR.** This paper investigates whether the attribution map trends observed in the super-resolution model PASD Yang et al. (2023b) are also present in the SR model SeeSR Wu et al. (2024). The SR models discussed here provide low-resolution signals during inference. As mentioned in previous works (Choi et al., 2022; Lin et al., 2024), the noise schedule used in the representative latent diffusion model, Stable Diffusion, does not guarantee an SNR (signal-

to-noise ratio) of zero at the final time step during training, leading to residual signals and train-test discrepancies. SeeSR and PASD combine low-quality (LQ) input with pure noise during inference to address the mismatch. Specifically, when independent Gaussian noises $\mathbf{z}_T$ and $\mathbf{z}'_T$ are present, conventional inference starts the denoising process from $\mathbf{z}_T$, whereas SeeSR and PASD proceed as follows:

$$\mathbf{z}_T^{\text{seesr}} = \sqrt{\bar{\alpha}_T}\mathbf{z}_{\text{LR}} + \sqrt{1 - \bar{\alpha}_T}\mathbf{z}_T \tag{24}$$

$$\mathbf{z}_T^{\text{pasd}} = \sqrt{\bar{\alpha}_a\bar{\alpha}_T}\mathbf{z}_{\text{LR}} + \sqrt{1 - \bar{\alpha}_a\bar{\alpha}_T}\mathbf{z}'_T \tag{25}$$

where $T$ is the terminal time step, $\mathbf{z}_{\text{LR}}$ is the LQ latent, $\bar{\alpha}_T$ is the noise magnitude, and $\bar{\alpha}_a$ controls the residual signal $\mathbf{z}_{\text{LR}}$. In PASD, $\bar{\alpha}_a$ is set to $0.1189$. Therefore, both PASD and SeeSR, as experimented with in this paper, receive low-frequency information along with the input, allowing the initial layers to remain less influenced by coarse features, as shown in 2a.

While PASD introduces $\bar{\alpha}_a$ to reduce excessive interference from $\mathbf{z}_{\text{LQ}}$, SeeSR relies on a higher volume of LQ data. As shown in Fig. 17, our visualization method highlights the differences in generation processes based on initial noise. Fig. 17 presents SeeSR results for the prompt "`close-up, electronic, clean, high-resolution, 8k`" on the same sample. SeeSR uses different numbers of attention heads depending on resolution (16 resolution: 20 heads, 32 resolution: 10 heads, 64 resolution: 5 heads), totaling 35 attention heads, with 8, 4, and 2 heads extracted from 16, 32, and 64 resolutions, respectively, for visualization. Additionally, SeeSR uses 50 time steps ($T_{\text{group}} = 5$) compared to PASD's 20 ($T_{\text{group}} = 4$), leading to longer generation times. The narrower attention score range in SeeSR likely reflects either distortion due to strong LQ signals or sufficient information being provided. HLAMs ($\mathbf{M}_{\text{g},n}, \mathbf{M}_{\text{r},n}$) show that each attention head learns different information tailored to diverse parts of the input image, similar to PASD, and this trend is also observed in TLAM $\mathbf{M}_{\text{g},\tau}$. In TLAM $\mathbf{M}_{\text{r},\tau}$, differences in LQ input levels during inference impact visualization and lead to less consistent attention scores. This indicates that SeeSR's longer time steps and stronger LQ signals offer more information early in the process, with consistency decreasing over time. Similarly, as shown in Fig. 18, we observe that low involvement in coarse features of the early layers and positional alignment weakens from low-resolution to high-resolution layers through SRAM $\mathbf{M}_{\text{sr},i}^{(l)}$, following the same trend as PASD.

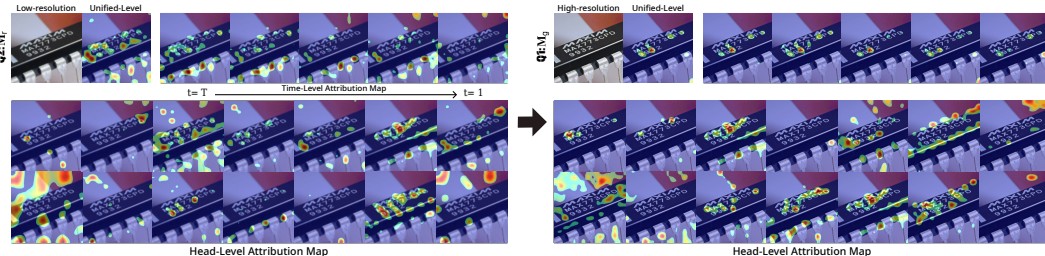

Figure 17: Visualization of SeeSR super-resolution results across 35 attention heads: 8 heads at 16 resolution, 4 heads at 32 resolution, and 2 heads at 64 resolution. The prompt used for this visualization is "`close-up, electronic, clean, high-resolution, 8k`". HLAMs ($\mathbf{M}_{\text{g},n}, \mathbf{M}_{\text{r},n}$) show each head learning and adapting to different parts of the input image. $\mathbf{M}_{\text{g},\tau}$ aligns with PASD patterns, while $\mathbf{M}_{\text{r},\tau}$ displays less consistency in attention scores over time, likely due to SeeSR's longer time steps and stronger low-quality signals. The prominent low-quality signal also results in a narrower distribution of attention scores.

**Insights into SRAM.** SRAM provides mapping information between each patch of the generated image and the reference image, addressing distortions caused by unnecessary patches. In Fig. 1, ULAM $\mathbf{M}_{\text{g}}$ effectively uses reference image data, while ULAM $\mathbf{M}_{\text{r}}$ struggles to extract critical details, such as clothing logos, leading to potential detail loss. This distortion arises from the G2R attribution map, which treats the generated image as a single token. SRAM, however, provides attribution maps for each patch (e.g., the red box around the logo), confirming that semantic information is accurately extracted from the reference patches. Notably, the first row, the second column of the grid highlights the retrieval of logo details from the reference image, ensuring precise logo generation.

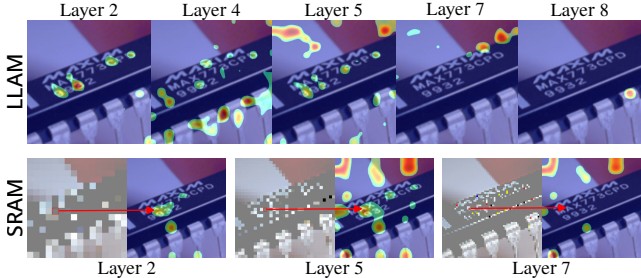

Figure 18: LLAM $\mathbf{M}_{\mathsf{g}}^{(l)}$: Strong low-quality signal in initial noise results in lower attention score on coarse features; SRAM $\mathbf{M}_{\mathsf{sr},i}^{(l)}$: Lower layers exhibit geometric correspondence, but this trend diminishes as the resolution increases.

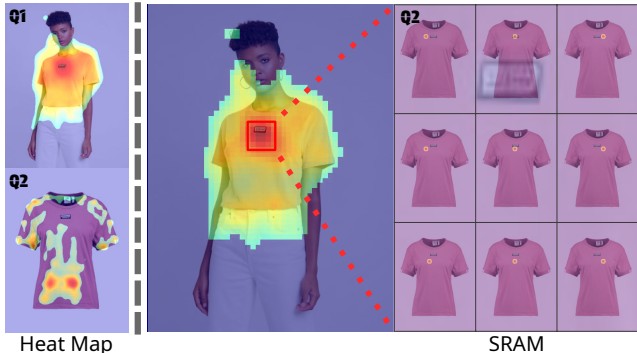

Figure 19: SRAM $\mathbf{M}_{\mathsf{sr},i}^{(l)}$ shows the alignment between the specific generated patch and reference image, enabling verification of effective information transfer. This figure, an enlarged view of Fig 1, illustrates the concept. While the bottom map in the first column appears to miss critical details (e.g., a clothing logo), SRAM reveals that the generated patches in the red box, including the logo area, are correctly matched to regions of the reference image.

## A.8 LIMITATION

We calculate the attention map by considering the characteristics of the diffusion model, segmenting it by time step $t$, attention heads $n$, and layer $l$. Based on this, we propose ULAM, which combines all axes, as well as TLAM (segmented by time step), HLAM (segmented by attention head), and LLAM (segmented by layer). However, it is also possible to calculate the map by considering two axes simultaneously, rather than just one. This can be achieved by modifying the aggregation in equations 11, 13, and 14. For example, as shown in Fig. 2, we can examine the roles of layers at different resolutions in the inpainting model, considering both the layer and other axes. The results are shown in Fig. 20.

Fig. 20a shows the TLAM for each layer at different resolutions (16, 32, 64), where we observe minor changes in attention scores, but the result is similar to the original TLAM $\mathbf{M}_{\mathsf{g},\tau}$. Fig. 20b presents the HLAM for each layer at different resolutions, highlighting the varying distribution of attention scores across heads for each layer. However, since we expected the HLAM to capture the extraction of information from diverse regions by each head, this characteristic is sufficiently confirmed through the original HLAM $\mathbf{M}_{\mathsf{g},n}$, which integrates all layers.

We conducted experiments in a paired setting. Previous study Jia et al. (2021) has shown that in an unpaired setting, image-to-image translation leads to semantic flip to maintain the distribution of transformed images, often reversing the meaning of the input despite visually plausible outputs. Although we do not provide a deep analysis of the unpaired setting, the results of StableVITON in the unpaired setting, shown in Fig. 21, maintain similar consistency to the paired setting. This suggests the potential to extend $\mathrm{I}^2\mathrm{AM}$ to unpaired setups. Finally, while we intended to apply our method to tasks like colorization, depth estimation, and style transfer, due to the lack of models that use cross-attention with reference images, we only present results for object detection, image

inpainting, and super-resolution. Visualization of attribution maps for more tasks remains a direction for future work.

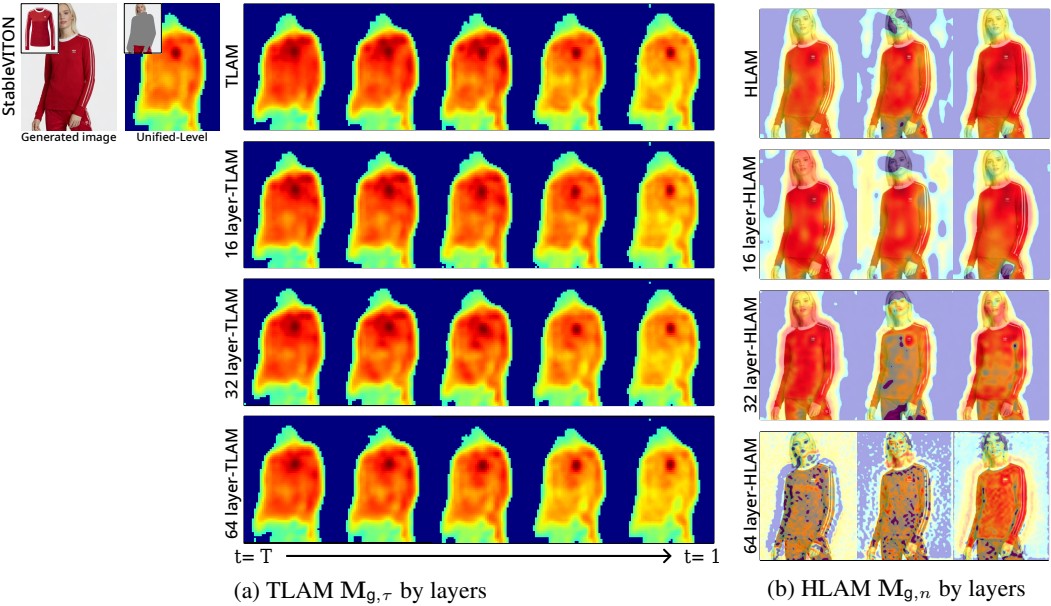

(a) TLAM $\mathbf{M}_{g,\tau}$ by layers   (b) HLAM $\mathbf{M}_{g,n}$ by layers

Figure 20: The R2G attribution map aggregated by layer resolution is shown. It consists of 9 layers grouped into three sets with 16, 32, and 64 resolutions. (a) Compared to TLAM, the attention scores differ in detail but maintain the same overall pattern over time. (b) The HLAM shows 3 of the 8 heads. Each layer has a different attention distribution, and even the same head index does not focus on the same perspective. This observation indicates that each head attends to different regions, allowing the model to process diverse types of information effectively.

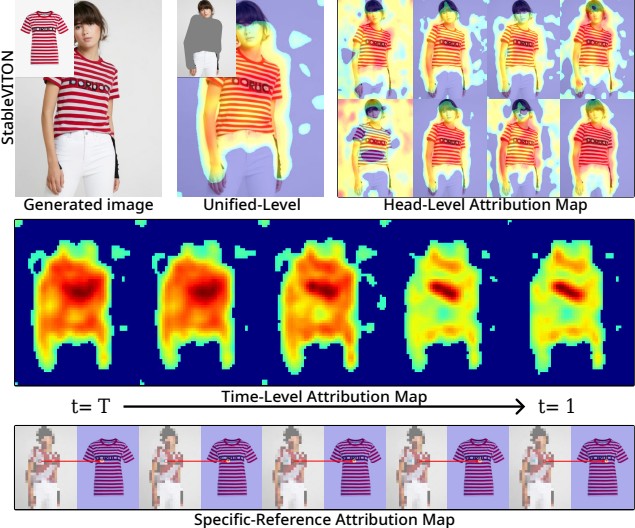

Figure 21: Visualization results of $I^2$AM in an unpaired setting. The attribution map shows similar trends to the paired setting, suggesting the model's robustness in the unpaired environment. ULAM $\mathbf{M}_g$: A map that reveals the overall trend, with scores evenly distributed within the masked region. HLAM $\mathbf{M}_{g,n}$: Each head exhibits rich expressiveness and considers global relationships, even in the unpaired setting. TLAM $\mathbf{M}_{g,\tau}$: Detailed attention scores are consistently maintained. SRAM $\mathbf{M}_{sr,i}^{(l)}$: Mapping information for generated patches works correctly in the unpaired setting.

