# OpenReview forum: "$\text{I}^2\text{AM}$: Interpreting Image-to-Image Latent Diffusion Models via Bi-Attribution Maps"
_ICLR.cc/2025/Conference — ICLR 2025 Poster_

### Official Review · Reviewer_bmkU · 2024-10-27

**Soundness:** 2
**Presentation:** 2
**Contribution:** 2
**Rating:** 5
**Confidence:** 5

**Summary:**

The paper proposes a method called Image-to-Image Attention Mapping (I2AM) for improving the explainability of diffusion models. I2AM visualizes bidirectional attention mapping between reference and generated images to understand how the model processes image data. This method is particularly useful for image-to-image (I2I) tasks since it can capture complex relationships between reference and generated images. Furthermore, the authors introduce a new metric called Image Repair Attention Consistency Score (IMACS) to evaluate model performance and attention distribution consistency

**Strengths:**

1. The proposed method offers a novel perspective on interpreting LDMs by focusing on the relationship between reference and generated images.

2. The authors also introduce a new metric, Image Repair Attention Consistency Score (IMACS), to evaluate model performance and attention distribution consistency, which adds value to the research.

3. The experimental results demonstrate the effectiveness of I2AM in identifying key feature regions for tasks such as object detection, image repair, and super-resolution.

4. The paper is written in a clear and concise manner.

**Weaknesses:**

1. The paper does not include an ablation study to demonstrate the effectiveness of the core components of the proposed method.

2. The authors compare their method with several baseline models, but they do not mention any other attention-based methods or explain why they chose those specific models. Adding comparisons with other attention-based methods would help to understand the relative strengths and weaknesses of the proposed approach.

3. The authors do not discuss the limitations of their method.  Does the method work well for any diffusion models?

**Questions:**

See Weaknesses.  I will change my rating based on the responses.

---

> ### Author Response · Authors · 2024-11-22
> **Response to Reviewer bmkU (1/3)**
>
> We sincerely appreciate the reviewer’s valuable feedback. Based on the provided comments, we have updated the experimental results and paper, with the changes highlighted in blue. If you have any further concerns, please feel free to let us know. We hope the responses are clear and kindly request that you consider a higher evaluation score.
>
> **W1. The paper does not include an ablation study to demonstrate the effectiveness of the core components of the proposed method.**
> > - Thank you for emphasizing the importance of an ablation study. We have addressed this concern by including a detailed ablation study in **Appendix A.6**. This study specifically justifies our selection of $\\delta$ and $\\lambda$, which represent the threshold of attention maps and the penalty factor in IMACS, respectively. We believe this analysis demonstrates how these parameters contribute to the overall performance and interpretability of our approach.
> > - In **Table 3**, we evaluated the impact of varying $\delta$ on the object detection task. The results demonstrate that our chosen value effectively balances precision and recall, validating its effectiveness. In **Table 4**, we analyzed the effect of varying $\\lambda$ in IMACS. We observed that the results remained consistent with downstream task metrics. However, models with significant information loss (e.g., DCI-VTON) exhibited sharp changes in scores as $\\lambda$ increased. This sensitivity highlights the importance of $\\lambda$ in penalizing attention misalignment effectively.
>
> $$
> \\begin{array}{lrrrrr}
> \\hline
>  & &  \\text{mIOU}\_{\\text{gen}}\^{>0.5} & & &
> \\\\
> \\text{Method} & \\delta=0.2 & \\delta=0.3 & \\delta=0.4 & \\delta=0.5 & \\delta=0.6
> \\\\
> \\text{Ours} & 0.2413 & 0.2413 & \\textbf{0.2416} & \\textbf{0.2416} & 0.24
> \\\\ \\hline
> \\end{array}
> $$
>
> $$
> \\begin{array}{l|rr|rr|rr|rr}
> \\hline
>  & \\lambda=1 & & \\lambda=3 & & \\lambda=5 & (\text{default}) & \\lambda=10 &
> \\\\
> \text{Method} & \\text{IMACS}\_{\\textsf{g}}  \\uparrow &  \\text{IMACS}\_{\\textsf{r}}   \\uparrow & \\text{IMACS}\_{\\textsf{g}}   & \\text{IMACS}\_{\\textsf{r}}  & \\text{IMACS}\_{\\textsf{g}}  &  \\text{IMACS}\_{\\textsf{r}}  & \\text{IMACS}\_{\\textsf{g}}  &  \\text{IMACS}\_{\\textsf{r}}
> \\\\ \\hline
> \text{DCI-VTON} & 0.4291 & - & 0.0785 & - & -0.3051 & - & -1.2155 & - \\\\
> \text{StableVITON} & \\textbf{0.5376} & \\textbf{0.3534} & \\textbf{0.3083} & \\textbf{0.3388} & \\textbf{0.07} & \\textbf{0.3211} & \\textbf{-0.1219} & \\textbf{0.301} \\\\
> \text{Custom} & 0.2352 & 0.0677 & 0.0833 & 0.0403 & -0.1901 & 0.0091 & -0.4579 & -0.0527 \\\\
> \text{Refined custom} & \\underline{0.4675} & \\underline{0.1249} & \\underline{0.2215} & \\underline{0.0948} & \\underline{0.029}
> & \\underline{0.0513} & \\underline{-0.3105} & \\underline{0.0112} \\\\ \\hline
> \\end{array}
> $$

---

> ### Author Response · Authors · 2024-11-26
> **Response to Reviewer bmkU (2/3)**
>
> **W2. The authors compare their method with several baseline models, but they do not mention any other attention-based methods or explain why they chose those specific models. Adding comparisons with other attention-based methods would help to understand the relative strengths and weaknesses of the proposed approach.**
> > - Thank you for your valuable feedback. In response to your comment, we have added a comparison with another attention-based attribution method, DAAM [15]. This comparison is shown in **Tab. 1** (below), and we provide a more detailed discussion in the **Related work (L121-L125) and Appendix (A.7: Comparison with DAAM)**.
> >    1. **Key differences with T2I visualization** (e.g., DAAM [15]): Unlike methods like DAAM, which focuses on text-to-image models, our approach addresses the unique characteristics of I2I tasks. DAAM applies softmax in a single direction (query: generated image, key: textual tokens), which is unsuitable for reference images that maintain spatial and contextual continuity. Given this continuity, the reference image should be considered as a single token in DAAM approach, and thus applying the same operation in DAAM leads to uniform attention scores across all patches in the reference image, limiting its interpretability.
> >    2. **Bidirectional attribution maps**: We propose two attribution maps (as described in **eq(6)**) - R2G ($\\mathbf{M}\_{\\textsf{g},t,n}\^{(l)}$) and G2R ($\\mathbf{M}\_{\\textsf{r},t,n}\^{(l)}$), which address the bidirectional relationship between the reference and generated images. This approach allows us to visualize how patches from the reference image influence those in the generated image and vice versa, overcoming the limitations of treating the reference image as a single token. With these two maps, we leverage the reference image’s spatial continuity to preserve fine-grained details. The R2G map (query: reference image, key: generated image) enables the visualization of patch-level contributions in the generated images, even with only the CLS embedding. This method provides insights that traditional softmax operations used in T2I tasks cannot offer. As shown in **Tab. 1** (below), applying DAAM to paint-by-example (PBE) generates the same map as **Fig. 16(b)**, resulting in identical values for "overall.”
> >    3. **Novelty and Research Potential**: Existing latent diffusion models (LDMs) typically process reference images by simply concatenating or adding them to the input images [1, 7, 8, 9, 13], which limits their interpretability. In contrast, the cross-attention mechanism utilized in this work independently learns the relationships between the reference and input images, emphasizing their interactions and enhancing interpretability. Recent methods that utilize cross-attention [2, 3, 4, 5, 6, 10, 11, 12, 14] have gained attention for their effectiveness in visualizing the interactions between reference images and generated elements by using $\text{I}^2\text{AM}$, demonstrating great potential for advancing I2I tasks.
>
> $$
> \\begin{array}{lr}
> \\hline
>  \\text{Unsupervised manner} & \\& \\text{Unseen dataset}
> \\\\ \\hline
> \\text{DAAM} & 0.1807 \\\\
> \\text{Overall} & 0.1807 \\\\
> \text{Random} & 0.2028
> \\\\ \\hline
> \\text{Ours} & 0.2416
>  \\\\ \\hline
> \\end{array}
> $$

---

> ### Author Response · Authors · 2024-11-26
> **Reference for W2.**
>
> [1] Zhang, Lvmin, Anyi Rao, and Maneesh Agrawala. "Adding conditional control to text-to-image diffusion models." *Proceedings of the IEEE/CVF International Conference on Computer Vision*. 2023.
>
> [2] Yang, Binxin, et al. "Paint by example: Exemplar-based image editing with diffusion models." *Proceedings of the IEEE/CVF Conference on Computer Vision and Pattern Recognition*. 2023.
>
> [3] Kim, Jeongho, et al. "Stableviton: Learning semantic correspondence with latent diffusion model for virtual try-on." *Proceedings of the IEEE/CVF Conference on Computer Vision and Pattern Recognition*. 2024.
>
> [4] Yang, Tao, et al. "Pixel-aware stable diffusion for realistic image super-resolution and personalized stylization." *arXiv preprint arXiv:2308.14469* (2023).
>
> [5] Wu, Rongyuan, et al. "Seesr: Towards semantics-aware real-world image super-resolution." *Proceedings of the IEEE/CVF conference on computer vision and pattern recognition*. 2024.
>
> [6]Kim, Kangyeol, et al. "Reference-based image composition with sketch via structure-aware diffusion model." *arXiv preprint arXiv:2304.09748* (2023).
>
> [7] Saharia, Chitwan, et al. "Image super-resolution via iterative refinement." *IEEE transactions on pattern analysis and machine intelligence* 45.4 (2022): 4713-4726.
>
> [8] Meng, Chenlin, et al. "Sdedit: Guided image synthesis and editing with stochastic differential equations." *arXiv preprint arXiv:2108.01073* (2021).
>
> [9] Choi, Jooyoung, et al. "Ilvr: Conditioning method for denoising diffusion probabilistic models." *arXiv preprint arXiv:2108.02938* (2021).
>
> [10] Wang, Qixun, et al. "Instantid: Zero-shot identity-preserving generation in seconds." *arXiv preprint arXiv:2401.07519* (2024).
>
> [11] Ye, Hu, et al. "Ip-adapter: Text compatible image prompt adapter for text-to-image diffusion models." *arXiv preprint arXiv:2308.06721* (2023).
>
> [12] Wang, Chenhui, et al. "FLDM-VTON: Faithful Latent Diffusion Model for Virtual Try-on." *arXiv preprint arXiv:2404.14162* (2024).
>
> [13] Li, Leheng, et al. "OmniBooth: Learning Latent Control for Image Synthesis with Multi-modal Instruction." *arXiv preprint arXiv:2410.04932* (2024).
>
> [14] Seyfioglu, Mehmet Saygin, et al. "Diffuse to Choose: Enriching Image
> Conditioned Inpainting in Latent Diffusion Models for Virtual Try-All." *arXiv preprint arXiv:2401.13795* (2024).
>
> [15] Tang, Raphael, et al. "What the daam: Interpreting stable diffusion using cross attention." *arXiv preprint arXiv:2210.04885* (2022).

---

> ### Author Response · Authors · 2024-11-26
> **Response to Reviewer bmkU (3/3)**
>
> **W3. The authors do not discuss the limitations of their method. Does the method work well for any diffusion models?**
> > - Thank you for your insightful comment. We recognize the importance of discussing the limitations of our method and have included a detailed analysis, along with future opportunities, in both the **Conclusion and Appendix A.8**. Here, we summarize the key limitations:
> >   1. **Limited axis analysis**: Although we focused on analyzing bidirectional maps on each axis (time step, attention heads, and cross-attention layers) individually, we did not explore the simultaneous analysis of multiple axes in detail. To address this, we have added new results in **Fig.20** that investigate both head-layer and time-layer axes simultaneously, providing a more comprehensive analysis.
> >   2. **Paired setting experiments**: Our initial experiments were focused on a paired setting. We did not initially investigate the unpaired setting, which can introduce semantic mismatches or flips, potentially leading to undesired results [1]. In the revised manuscript, we conducted additional experiments in an unpaired setup (see **Fig.21**) and found that our bidirectional attribution maps still exhibit similar trends, indicating their robustness in this context.
> >   3. **Task and model scope** (*Response to "Does the method work well for any diffusion models?"*): While our method effectively applies to various latent diffusion models, its applicability is limited by specific structural requirements:
> >      - *i) The reference image must convey information through a cross-attention mechanism*
> >      - *ii) complete patch embeddings of the reference image must be available to compute the attribution maps for reference image*.
> > - These limitations constrained the scope of our experiments to tasks and models that meet these criteria. However, as discussed in **response to Reviewer HMWz’s W2**, we anticipate that the scalability of our method will improve as cross-attention mechanisms in LDMs that support reference images become more common in future models.
>
> [1] Semantically Robust Unpaired Image Translation for Data With Unmatched Semantics Statistics, ICCV 2021

---

> ### Author Response · Authors · 2024-12-02
> **Official Comment by Reviewer bmkU**
>
> **Dear Reviewer bmkU**,
>
> As the ICLR discussion phase is nearing its conclusion, we would like to gently remind you to review our responses to the comments and questions you raised. Your feedback is highly valued and will be crucial in refining our work. We would greatly appreciate any additional thoughts or suggestions you may have.
>
> Thank you again for your time and valuable insights throughout the review process. We look forward to your response.
>
> Best regards,
> *Authors of Submission 5933*

---

### Official Review · Reviewer_N4o3 · 2024-10-28

**Soundness:** 3
**Presentation:** 3
**Contribution:** 3
**Rating:** 6
**Confidence:** 3

**Summary:**

This paper presents a method to improve the interpretability of Image-to-Image (I2I) models by visualizing bidirectional attribution maps and proposing a new evaluation metric. The introduced Image-to-Image Attribution Maps ($I^2AM$) aggregate cross-attention scores across time steps, attention heads, and layers, providing insights into how key features transfer between images. Experimental results indicate that this method, enables model debugging and refinement, offering practical tools to enhance I2I model performance and interpretability.

**Strengths:**

- The paper introduces a well-designed method that improves interpretability in I2I models.

- The analysis of Image-to-Image Attribution Maps is interesting, demonstrating the utility of $I^2AM$ across tasks like object detection, inpainting, and super-resolution.

- The writing and presentation are generally clear and accessible, making it easier for readers to follow the proposed method.

**Weaknesses:**

- In the "Unsupervised manner & Unseen dataset" section of Table 1, a baseline comparison is missing. Applying [1]'s attribute map to identify key regions would provide a fair comparison and strengthen the analysis.

- Additional comparative experiments and visualizations would benefit the paper. For instance, more qualitative examples in tasks such as object detection, image inpainting, and super-resolution would enhance understanding of the method's effectiveness.

- The Specific-reference Attribution Map (SRAM) lacks formal definitions. It remains unclear how reference patch selection and attention aggregation are conducted, even with the explanations in Figure 4.

-  The choices for parameters δ and λ are not adequately justified through experiments. Additional ablation studies or experiments exploring the effect of varying δ and λ values would strengthen the argument for their selection, making the methodology more transparent and grounded.

[1] Tang, Raphael, et al. “What the DAAM: Interpreting Stable Diffusion Using Cross Attention.” Annual Meeting of the Association for Computational Linguistics (2022).

**Questions:**

Please refer to the Weaknesses for more details.

---

> ### Author Response · Authors · 2024-11-22
> **Response to Reviewer N4o3 (1/2)**
>
> We sincerely appreciate the reviewer’s valuable feedback. Based on the provided comments, we have updated the experimental results and paper, with the changes highlighted in blue. If you have any further concerns, please feel free to let us know. We hope the responses are clear and kindly request that you consider a higher evaluation score.
>
> **W1. In the "Unsupervised manner & Unseen dataset" section of Table 1, a baseline comparison is missing. Applying [1]'s attribute map to identify key regions would provide a fair comparison and strengthen the analysis.**
> > - Thank you for pointing out the need for a baseline comparison in **Table 1**. We have added a detailed comparison with DAAM [1] in **Table 1** and have expanded this discussion in **Appendix (A.7: Comparison with DAAM)**. This information is also reflected in our responses to **Reviewer 4XJm, W1**.
> > - As shown in the revised **Table 1 (below)**, DAAM achieves matching “overall” values for the object detection task. This result is due to the differences in the operation of Softmax between DAAM and our method. In I2I tasks, DAAM’s approach uses the reference image for conditioning, which is characterized by contextual continuity. This makes token-by-token interpretation through DAAM less meaningful, as the reference image needs to be treated as a single token. This often leads to identical attention scores, which limits the effectiveness of visualizing fine-grained contributions from the reference image. Qualitative results illustrating this limitation are included in **Fig.16**, where we demonstrate how our method addresses these challenges and provides more interpretable visualizations.
>
> $$
> \\begin{array}{lr}
> \\hline
>  \\text{Unsupervised manner} & \\& \\text{Unseen dataset}
> \\\\ \\hline
> \\text{DAAM} & 0.1807 \\\\
> \\text{Overall} & 0.1807 \\\\
> \text{Random} & 0.2028
> \\\\ \\hline
> \\text{Ours} & 0.2416
>  \\\\ \\hline
> \\end{array}
> $$
>
> **W2. Additional comparative experiments and visualizations would benefit the paper. For instance, more qualitative examples in tasks such as object detection, image inpainting, and super-resolution would enhance understanding of the method's effectiveness.**
> > - We appreciate the reviewers’ suggestion to conduct further experiments to better demonstrate the effectiveness of our method, which have been added as follows:
> >   1. **Super-resolution task**: We have included experiments using the recent model SeeSR [1], with results presented in **Fig.18** and **Fig.19**. By visualizing the same samples used with PASD, we confirmed that SeeSR shows similar trends in attention behavior, thereby further validating our findings across different super-resolution models.
> >   2. **Image inpainting task**: Additional qualitative examples for inpainting tasks are provided in **Fig.20** and **Fig.21**. These visualizations highlight TLAM and HLAM at various layer resolutions and evaluate consistency in trends under unpaired settings.
>
> [1] Wu, Rongyuan, et al. "Seesr: Towards semantics-aware real-world image super-resolution." Proceedings of the IEEE/CVF conference on computer vision and pattern recognition. 2024.

---

> ### Author Response · Authors · 2024-11-22
> **Response to Reviewer N4o3 (2/2)**
>
> **W3. The Specific-reference Attribution Map (SRAM) lacks formal definitions. It remains unclear how reference patch selection and attention aggregation are conducted, even with the explanations in Figure 4.**
> > - We have provided a detailed formal definition of SRAM in **Appendix A.3 under “Specific-Reference Attribution Map”**. To clarify the computation process, we outline the following steps:
> >   - We first extract $\\mathbf{M}\_{\\textsf{r},t,n}\^{(l)}$ from the desired layer $l$ using **eq (6)**. This matrix represents the relationship between each generated patch and the reference image.
> >   - Then, to compute the SRAM $\\mathbf{M}\_{\\textsf{sr},i}\^{(l)}$, we select a specific generated patch $i$ and aggregate the attention scores over the time step $t$ and attention head $n$. This process identifies which regions of the reference image that contribute to or align with the selected generated patch $i$. The computation is expressed as follows:
> >   - $\\mathbf{M}\_{\\textsf{sr},i}\^{(l)}  = \\text{Sum}\_{\\{t,n\\}}(\\mathbf{M}\_{\\textsf{r},t,n}\^{(l)}[i,:])
> $
> >   - The SRAM obtained through this process highlights the reference regions associated with a specific generated patch, such as the logo on clothing, as shown in **Fig.4**. By analyzing these attributions, we can tell which areas of the reference image are most significant for generating specific details. Additionally, we note that SRAM can be calculated for all generated patches, providing a comprehensive understanding of how different patches in the reference image contribute to the entire generation process.
>
> **W4. The choices for parameters δ and λ are not adequately justified through experiments. Additional ablation studies or experiments exploring the effect of varying δ and λ values would strengthen the argument for their selection, making the methodology more transparent and grounded.**
> > - Thank you for your valuable feedback. To address your concerns, we conducted an ablation study, which is now included in **Appendix A.6**. This study justifies our selection of $\\delta$ and $\\lambda$, representing the threshold of attention maps and penalty factor in IMACS, respectively.
> > - In **Table 3**, we evaluated the impact of varying $\delta$ on the object detection task. The results demonstrate that our chosen value effectively balances precision and recall, validating its effectiveness. In **Table 4**, we analyzed the effect of varying $\\lambda$ in IMACS. We observed that the results remained consistent with downstream task metrics. However, models with significant information loss (e.g., DCI-VTON) exhibited sharp changes in scores as $\\lambda$ increased. This sensitivity highlights the importance of $\\lambda$ in penalizing attention misalignment effectively.
>
> $$
> \\begin{array}{lrrrrr}
> \\hline
>  & &  \\text{mIOU}\_{\\text{gen}}\^{>0.5} & & &
> \\\\
> \\text{Method} & \\delta=0.2 & \\delta=0.3 & \\delta=0.4 & \\delta=0.5 & \\delta=0.6
> \\\\
> \\text{Ours} & 0.2413 & 0.2413 & \\textbf{0.2416} & \\textbf{0.2416} & 0.24
> \\\\ \\hline
> \\end{array}
> $$
>
> $$
> \\begin{array}{l|rr|rr|rr|rr}
> \\hline
>  & \\lambda=1 & & \\lambda=3 & & \\lambda=5 & (\text{default}) & \\lambda=10 &
> \\\\
> \text{Method} & \\text{IMACS}\_{\\textsf{g}}  \\uparrow &  \\text{IMACS}\_{\\textsf{r}}   \\uparrow & \\text{IMACS}\_{\\textsf{g}}   & \\text{IMACS}\_{\\textsf{r}}  & \\text{IMACS}\_{\\textsf{g}}  &  \\text{IMACS}\_{\\textsf{r}}  & \\text{IMACS}\_{\\textsf{g}}  &  \\text{IMACS}\_{\\textsf{r}}
> \\\\ \\hline
> \text{DCI-VTON} & 0.4291 & - & 0.0785 & - & -0.3051 & - & -1.2155 & - \\\\
> \text{StableVITON} & \\textbf{0.5376} & \\textbf{0.3534} & \\textbf{0.3083} & \\textbf{0.3388} & \\textbf{0.07} & \\textbf{0.3211} & \\textbf{-0.1219} & \\textbf{0.301} \\\\
> \text{Custom} & 0.2352 & 0.0677 & 0.0833 & 0.0403 & -0.1901 & 0.0091 & -0.4579 & -0.0527 \\\\
> \text{Refined custom} & \\underline{0.4675} & \\underline{0.1249} & \\underline{0.2215} & \\underline{0.0948} & \\underline{0.029}
> & \\underline{0.0513} & \\underline{-0.3105} & \\underline{0.0112} \\\\ \\hline
> \\end{array}
> $$

---

> > ### Comment · Reviewer_N4o3 · 2024-11-26
> >
> > Thank you for the authors’ response. Most of my concerns have been addressed, and I will maintain my current rating.

---

> > > ### Author Response · Authors · 2024-11-26
> > > **To Reviewer N4o3**
> > >
> > > Thank you for your feedback and for taking the time to review our response. We appreciate your consideration and are glad to have addressed your concerns. Please let us know if there are any additional points you’d like us to clarify or elaborate on.

---

### Official Review · Reviewer_zNnp · 2024-10-30

**Soundness:** 3
**Presentation:** 3
**Contribution:** 2
**Rating:** 6
**Confidence:** 3

**Summary:**

The interpretability of cross-attention mechanisms in image-to-image (I2I) diffusion models remains underexplored, while cross-attention maps in text-to-image (T2I) models have been extensively studied. This paper advances interpretability in I2I models by visualizing attributions from two perspectives: from the reference image to the generated image and from the generated image back to the reference. The authors propose a novel method for interpreting latent diffusion models using cross-attention maps from these dual perspectives. This approach enables researchers to debug and enhance I2I models in various tasks, such as object detection, inpainting, and super-resolution. Additionally, they introduce an inpainting mask attention consistency score as an evaluation metric.

**Strengths:**

- This method visualizes bidirectional attribution maps across multiple axes, including diffusion time steps, attention heads, and layers, offering a comprehensive analysis. It also reveals insights into the coarse-to-fine process as layers deepen, enhancing interpretability.

- Additionally, the paper introduces a specific-reference attribution map tailored for inpainting tasks, helping guide the model to focus on relevant object parts while disregarding irrelevant background information from the reference image.

- The authors conduct comprehensive experiments across various tasks to validate the improvements enabled by these visualization techniques.

**Weaknesses:**

- In ULAM, the column-wise averaging limits the ability to visualize specific region-to-region mappings. For instance, in virtual try-on applications, it’s challenging to verify whether a logo in the generated image aligns with the logo in the reference image. According to the authors' visualization in Figure 1, the reference image’s logo appears to contribute less to the generated image, making detailed correspondence difficult to interpret.

- Additionally, a question arises regarding the visualization approach: given a CLIP image embedding, how can each token be accurately mapped back to the original reference image? It remains unclear whether using a fixed CLIP image embedding alone is sufficient for effective visualization.

- Lastly, some formatting issues were noted, such as Figure 3 appearing before Figure 2.

**Questions:**

My main concern is that it remains unclear whether using a fixed CLIP image embedding alone is sufficient for effective visualization, as demonstrated in Figure 1. I would appreciate if the authors could provide further clarification on this point, as it may influence my overall rating.

---

> ### Author Response · Authors · 2024-11-22
> **Response to Reviewer zNnp (1/2)**
>
> We sincerely appreciate the reviewer’s valuable feedback. Based on the provided comments, we have updated the experimental results and paper, with the changes highlighted in blue. If you have any further concerns, please feel free to let us know. We hope the responses are clear and kindly request that you consider a higher evaluation score.
>
> **W1. In ULAM, the column-wise averaging limits the ability to visualize specific region-to-region mappings. For instance, in virtual try-on applications, it’s challenging to verify whether a logo in the generated image aligns with the logo in the reference image. According to the authors' visualization in Figure 1, the reference image’s logo appears to contribute less to the generated image, making detailed correspondence difficult to interpret.**
> > - Thank you for your observation regarding the limitations of ULAM in visualizing specific region-to-region mappings, particularly in virtual try-on applications. We acknowledge that in **Fig.1**, the logo in the reference image seems to contribute less to the generated image, making detailed correspondences harder to interpret. This limitation arises because the ULAM for G2R $\\mathbf{M}\_{\\textsf{r}}$ aggregates the entire generated image as a single token, potentially introducing irrelevant information from other patches and distorting region-specific mappings.
> > - To address this issue, we intentionally introduced SRAM $\\mathbf{M}\_{\\textsf{sr},i}\^{(l)}$, which more effectively captures the mapping contributions between generated patches and the reference image. In **Fig.1**, the right map overlays SRAM with a red box around the patch near the clothing logo, providing a clearer alignment. Each map in the 3x3 grid corresponds to the SRAM of a specific generated patch, enabling precise region-to-region visualization. Unlike $\\mathbf{M}\_{\\textsf{r}}$, SRAM allows us to extract semantically aligned information for each generated patch, making detailed correspondences, such as logo alignment, easier to interpret.
> > - Additionally, we recognize that the small size of **Fig.1** may make it difficult to discern these details. To enhance clarity, we provide an enlarged visualization and further analysis in **Fig.19**, which better highlights these region-to-region mappings.
>
> **W2. Additionally, a question arises regarding the visualization approach: given a CLIP image embedding, how can each token be accurately mapped back to the original reference image? It remains unclear whether using a fixed CLIP image embedding alone is sufficient for effective visualization. & Q1. My main concern is that it remains unclear whether using a fixed CLIP image embedding alone is sufficient for effective visualization, as demonstrated in Figure 1. I would appreciate it if the authors could provide further clarification on this point, as it may influence my overall rating.**
> > - Thank you for raising the important issue regarding the use of fixed CLIP image embeddings for visualization. To effectively visualize attribution maps, it is essential to preserve spatial information of the reference image. This requires utilizing all patch embeddings in addition to the CLS embedding. In response to the reviewer’s question, a fixed CLIP image embedding can generally be utilized in two ways:
> >   1. **Using only the CLS embedding** (e.g., DCI-VTON, Paint-by-Example): In this approach, spatial information is not retained, which means that interpretability is limited to reference-to-generated (R2G) attention maps. While this is sufficient for certain applications, it fails to capture detailed region-to-region mappings due to the lack of spatial structure.
> >   2. **Using both CLS embedding and all patch embeddings** (e.g., custom model): This method preserves the spatial structure of the reference image, allowing bidirectional visualization. Both R2G and generated-to-reference (G2R) attention maps can be effectively computed, providing detailed insights into region-to-region mappings.
> > - The bidirectional visualization presented in **Fig.1** is based on StableVITON, following the structure outlined in **Fig.3(b)**, which incorporates spatial information from all patch embeddings. If the fixed CLIP image encoder supplies all patch embeddings, visualizations like those in **Fig.1** become achievable.
> > - For more details on embedding strategies and their implications for visualization, please refer to **Appendix A.4 (Implementation Details) for existing models and Appendix A.5 (Custom Model)**. These sections outline how spatial information is preserved in specific setups, supporting the interpretability demonstrated in our results.

---

> ### Author Response · Authors · 2024-11-22
> **Response to Reviewer zNnp (2/2)**
>
> **W3. Lastly, some formatting issues were noted, such as Figure 3 appearing before Figure 2.**
> > - Thank you for noting the formatting issue. We have corrected the figure order to ensure that they now appear in the sequence referenced in the main text.

---

> > ### Comment · Reviewer_zNnp · 2024-11-25
> >
> > Thanks for your response. I would like to improve my rating to 6

---

> > > ### Author Response · Authors · 2024-11-25
> > > **To Reviewer zNnp**
> > >
> > > We are very pleased to confirm that we have successfully addressed the major concerns raised by the reviewer. We would also like to express our sincere gratitude for the time and effort you dedicated to carefully reviewing our paper and providing valuable feedback. Your insights have been beneficial to our research and significantly improved the paper's quality. We will continue to reflect on areas for further enhancement and strive to present meaningful and comprehensive studies in the future.

---

### Official Review · Reviewer_4XJm · 2024-11-03

**Soundness:** 3
**Presentation:** 3
**Contribution:** 2
**Rating:** 6
**Confidence:** 3

**Summary:**

This paper describes  a way to aggregate and visualize cross-attention similarity maps in image-to-image latent diffusion models. Those visualizations help to understand which regions of the generated image were influenced most by the reference image, and which parts of the reference image contributed most to the generated image. The paper also proposes a new metric for evaluating reference-based image inpainting through estimating how well the inpainting model captures details from the reference image. They perform experiments on tasks like image inpainting and image super-resolution in order to show the visualization capabilities of the proposed method.

**Strengths:**

The proposed method for cross-attention similarity map visualizations is clearly explained.

**Weaknesses:**

1. The paper’s contribution appears incremental, focusing on the adaptation of existing methods to a new context. It mostly describes a way to aggregate and visualize cross-attention similarity maps of a pre-trained image-to-image diffusion denoiser network by averaging across different dimensions. Similar approaches have already been employed by other papers for visualizing the cross-attention layers, as also discussed in the related work of this paper. So the main novelty of this work is the application of the same principles of cross-attention map visualizations to the image-to-image diffusion models. Therefore, the significance of this work to the field is quite limited.

2. The head-level attribution map introduced in Section 4.3 may not be fully meaningful, as it averages across layers. It’s unclear why a given head index should play a consistent role across all layers or how its visualization provides useful information. The authors could consider clarifying this assumption.

3. Inpainting Mask Attention Similarity Score (IMACS) in Section 4.4, Equation 8, appears to yield a matrix, but it’s unclear how this is transformed into a scalar score. Additional clarification on this process would improve understanding.

4. Figure 2 needs more descriptive labeling. Terms such as c_I^{(l)}/c_I​ should be clearly defined within the figure caption or the main text, as their meaning is not immediately obvious.

5. Lines 304-305 state, “the layer-level maps offer a better indicator of model performance across different layers and help uncover areas for potential improvement.” It’s not clear how a layer-level attribution map can serve as a performance indicator. Additional explanation or evidence supporting this claim would be helpful.

**Questions:**

1. The "right map" in Figure 1 is difficult to interpret—could the authors clarify what it represents? Additionally, there appears to be an artifact in the first row, second column of the right 3x3 grid; what caused this?

2. In the experiment described in Section 5.4, what loss functions were used to "densify attention distributions" and "improve semantic alignment"? A description of these would be helpful.

---

> ### Author Response · Authors · 2024-11-22
> **Response to Reviewer 4XJm (1/3)**
>
> We sincerely appreciate the reviewer’s valuable feedback. Based on the provided comments, we have updated the experimental results and paper, with the changes highlighted in blue. If you have any further concerns, please feel free to let us know. We hope the responses are clear and kindly request that you consider a higher evaluation score.
>
> **W1. The paper's contribution appears incremental, mainly applying existing methods for visualizing cross-attention maps to image-to-image diffusion models. Since similar approaches have been used in previous works, the novelty and significance of this research to the field are limited.**
> > - We appreciate the opportunity to clarify our contributions. While our work builds on existing methods for visualizing cross-attention maps, it introduces significant innovations specifically tailored for image-to-image (I2I) diffusion models. We have outlined the differences with text-to-image (T2I) visualization methods in the **Related work (L121-L125) and Appendix (A.7: Comparison with DAAM)**.
> >    1. **Key differences with T2I visualization** (e.g., DAAM [15]): Unlike methods like DAAM, which focuses on text-to-image models, our approach addresses the unique characteristics of I2I tasks. DAAM applies softmax in a single direction (query: generated image, key: textual tokens), which is unsuitable for reference images that maintain spatial and contextual continuity. Given this continuity, the reference image should be considered as a single token in DAAM approach, and thus applying the same operation in DAAM leads to uniform attention scores across all patches in the reference image, limiting its interpretability.
> >    2. **Bidirectional attribution maps**: We propose two attribution maps (as described in **eq(6)**) - R2G ($\\mathbf{M}\_{\\textsf{g},t,n}\^{(l)}$) and G2R ($\\mathbf{M}\_{\\textsf{r},t,n}\^{(l)}$), which address the bidirectional relationship between the reference and generated images. This approach allows us to visualize how patches from the reference image influence those in the generated image and vice versa, overcoming the limitations of treating the reference image as a single token. With these two maps, we leverage the reference image’s spatial continuity to preserve fine-grained details. The R2G map (query: reference image, key: generated image) enables the visualization of patch-level contributions in the generated images, even with only the CLS embedding. This method provides insights that traditional softmax operations used in T2I tasks cannot offer. As shown in **Tab. 1** (below), applying DAAM to paint-by-example (PBE) generates the same map as **Fig. 16(b)**, resulting in identical values for "overall.”
> >    3. **Novelty and Research Potential**: Existing latent diffusion models (LDMs) typically process reference images by simply concatenating or adding them to the input images [1, 7, 8, 9, 13], which limits their interpretability. In contrast, the cross-attention mechanism utilized in this work independently learns the relationships between the reference and input images, emphasizing their interactions and enhancing interpretability. Recent methods that utilize cross-attention [2, 3, 4, 5, 6, 10, 11, 12, 14] have gained attention for their effectiveness in visualizing the interactions between reference images and generated elements by using $\text{I}^2\text{AM}$, demonstrating great potential for advancing I2I tasks.
>
> $$
> \\begin{array}{lr}
> \\hline
>  \\text{Unsupervised manner} & \\& \\text{Unseen dataset}
> \\\\ \\hline
> \\text{DAAM} & 0.1807 \\\\
> \\text{Overall} & 0.1807 \\\\
> \text{Random} & 0.2028
> \\\\ \\hline
> \\text{Ours} & 0.2416
>  \\\\ \\hline
> \\end{array}
> $$

---

> ### Author Response · Authors · 2024-11-22
> **Reference for W1.**
>
> [1] Zhang, Lvmin, Anyi Rao, and Maneesh Agrawala. "Adding conditional control to text-to-image diffusion models." *Proceedings of the IEEE/CVF International Conference on Computer Vision*. 2023.
>
> [2] Yang, Binxin, et al. "Paint by example: Exemplar-based image editing with diffusion models." *Proceedings of the IEEE/CVF Conference on Computer Vision and Pattern Recognition*. 2023.
>
> [3] Kim, Jeongho, et al. "Stableviton: Learning semantic correspondence with latent diffusion model for virtual try-on." *Proceedings of the IEEE/CVF Conference on Computer Vision and Pattern Recognition*. 2024.
>
> [4] Yang, Tao, et al. "Pixel-aware stable diffusion for realistic image super-resolution and personalized stylization." *arXiv preprint arXiv:2308.14469* (2023).
>
> [5] Wu, Rongyuan, et al. "Seesr: Towards semantics-aware real-world image super-resolution." *Proceedings of the IEEE/CVF conference on computer vision and pattern recognition*. 2024.
>
> [6]Kim, Kangyeol, et al. "Reference-based image composition with sketch via structure-aware diffusion model." *arXiv preprint arXiv:2304.09748* (2023).
>
> [7] Saharia, Chitwan, et al. "Image super-resolution via iterative refinement." *IEEE transactions on pattern analysis and machine intelligence* 45.4 (2022): 4713-4726.
>
> [8] Meng, Chenlin, et al. "Sdedit: Guided image synthesis and editing with stochastic differential equations." *arXiv preprint arXiv:2108.01073* (2021).
>
> [9] Choi, Jooyoung, et al. "Ilvr: Conditioning method for denoising diffusion probabilistic models." *arXiv preprint arXiv:2108.02938* (2021).
>
> [10] Wang, Qixun, et al. "Instantid: Zero-shot identity-preserving generation in seconds." *arXiv preprint arXiv:2401.07519* (2024).
>
> [11] Ye, Hu, et al. "Ip-adapter: Text compatible image prompt adapter for text-to-image diffusion models." *arXiv preprint arXiv:2308.06721* (2023).
>
> [12] Wang, Chenhui, et al. "FLDM-VTON: Faithful Latent Diffusion Model for Virtual Try-on." *arXiv preprint arXiv:2404.14162* (2024).
>
> [13] Li, Leheng, et al. "OmniBooth: Learning Latent Control for Image Synthesis with Multi-modal Instruction." *arXiv preprint arXiv:2410.04932* (2024).
>
> [14] Seyfioglu, Mehmet Saygin, et al. "Diffuse to Choose: Enriching Image
> Conditioned Inpainting in Latent Diffusion Models for Virtual Try-All." *arXiv preprint arXiv:2401.13795* (2024).
>
> [15] Tang, Raphael, et al. "What the daam: Interpreting stable diffusion using cross attention." *arXiv preprint arXiv:2210.04885* (2022).

---

> ### Author Response · Authors · 2024-11-22
> **Response to Reviewer 4XJm (2/3)**
>
> **W2. The head-level attribution map introduced in Section 4.3 may not be fully meaningful, as it averages across layers. It’s unclear why a given head index should play a consistent role across all layers or how its visualization provides useful information. The authors could consider clarifying this assumption.**
> > - We understand the reviewer’s observation that aggregating head-level attribution maps (HLAM) across all layers may obscure the unique roles of individual heads within specific layers. Initially, we assumed that each head captures distinct patterns or features, and our experimental results confirmed this expectation. While layer-wise aggregation may diminish the specific roles of heads in each layer, the aggregated HLAM still offers meaningful insights into the overall behavior of attention heads throughout the network.
> > - In response to the reviewer’s point, we conducted additional experiments, as illustrated in **Fig.17 and Fig.20 in the Appendix**. These visualizations clarify how attention heads function across different layers and complement the aggregated HLAM. In summary, we do not assume that a given head index maintains a consistent and fixed role across all layers. Instead, we acknowledge the variability in the roles of heads and provide both aggregated and layer-specific analysis to support various interpretive perspectives in this rebuttal.
>
> **W3. Figure 2 needs more descriptive labeling. Terms such as c_I^{(l)}/c_I should be clearly defined within the figure caption or the main text, as their meaning is not immediately obvious.**
> > - Thank you for highlighting the need for a clearer description in **Fig. 2**. We have updated the figure caption to explicitly define $c\_I\^{(l)}$ and $c\_I$, ensuring that their meaning is immediately understandable. Additionally, we clarified these terms in the main text on **L242-L244** to provide further context.
>
> **W4. Inpainting Mask Attention Similarity Score (IMACS) in Section 4.4, Equation 8, appears to yield a matrix, but it’s unclear how this is transformed into a scalar score. Additional clarification on this process would improve understanding.**
> > - We appreciate the reviewer’s thorough observation regarding the IMACS in **eq (8)**. As correctly noted, the result of **eq (8)** is indeed a matrix. This matrix is subsequently transformed into a scalar score by aggregating its values, which provides a single numerical measure of attention alignment. We have corrected **eq (8)** in the updated manuscript to clearly reflect this transformation process and included additional explanations detailing how the aggregation is performed. This ensures a better understanding of how IMACS yields a scalar score.
>
> $$\\text{IMACS}\_{\\textsf{g}} = \\frac{ \\sum\_{H,W}(\\mathbf{M}\_{\\textsf{g}} \\odot \\mathbf{x}\_{\\textsf{g}})}{\\sum_{H,W}\\mathbf{x}\_{\\textsf{g}}} - \\lambda \\frac{ \\sum\_{H,W}({\\mathbf{M}}\_{\\textsf{g}} \\odot (\\mathbf{1} - \\mathbf{x}\_{\\textsf{g}}))}{\\sum\_{H,W}(\\mathbf{1} - \\mathbf{x}\_{\\textsf{g}})},$$
>
> **W5. Lines 304-305 state, “The layer-level maps offer a better indicator of model performance across different layers and help uncover areas for potential improvement.” It’s not clear how a layer-level attribution map can serve as a performance indicator. Additional explanation or evidence supporting this claim would be helpful.**
> > - Thank you for raising this point and appreciate the opportunity to clarify the role of layer-level attribution maps (LLAMs) as performance indicators.
> > - Layer-level maps (e.g., attention map, feature map) capture diverse semantic information - such as object position, shape, and appearance - at different layers, as demonstrated in **Fig.2 and Fig.10**.  This information has been successfully utilized in various aspects, including optimization, image editing, and loss function design [1,2,3]. In our work, we examined specific characteristics of LLAMs and incorporated them into loss functions, which led to improved model performance (see **Fig. 10**).
> > - The effectiveness of LLAMs as performance indicators lies in their ability to represent the overall layer characteristics, in contrast to other types of attribution maps:
> >    1. *Head-level maps* are often seen to represent abstract relationships across attention heads, with each head corresponding to different pieces of information. Applying loss functions to these maps can disrupt the model’s learning process and potentially degrade performance.
> >    2. *Time-level maps* track incremental changes across diffusion steps, making it challenging to link these changes to overall model performance in general.
> > In summary, LLAMs provide a comprehensive view of each layer’s contribution, allowing loss functions to target specific improvements without interfering with the model’s natural learning dynamics. This makes them effective indicators for identifying areas where enhancements are possible.

---

> ### Author Response · Authors · 2024-11-22
> **Reference for W5.**
>
> [1] Shi, Yujun, et al. "Dragdiffusion: Harnessing diffusion models for interactive point-based image editing." *Proceedings of the IEEE/CVF Conference on Computer Vision and Pattern Recognition*. 2024.
>
> [2] Tumanyan, Narek, et al. "Plug-and-play diffusion features for text-driven image-to-image translation." *Proceedings of the IEEE/CVF Conference on Computer Vision and Pattern Recognition*. 2023.
>
> [3] Kim, Jeongho, et al. "Stableviton: Learning semantic correspondence with latent diffusion model for virtual try-on." *Proceedings of the IEEE/CVF Conference on Computer Vision and Pattern Recognition*. 2024.

---

> ### Author Response · Authors · 2024-11-22
> **Response to Reviewer 4XJm (3/3)**
>
> **Q1. The "right map" in Figure 1 is difficult to interpret—could the authors clarify what it represents? Additionally, there appears to be an artifact in the first row, second column of the right 3x3 grid; what caused this?**
> > - Thank you for pointing out the difficulties in understanding the right map in **Fig.1** and the confusing artifact. We have made the following clarifications:
> >   1. Interpretation of the right map in **Fig.1**: The “right map” in **Fig. 1** overlays the proposed SRAM with a red box highlighting the patch near the clothing logo. Each cell in the 3x3 grid represents the SRAM for a corresponding generated patch, indicating which reference information was used during its generation. To reduce ambiguity, we have revised the explanation in the main text (**L78-L80**):
> >   > “The right map illustrates which reference information was extracted during the generation of that cell.”
> >   2. Detailed analysis of **Fig. 1** for clarity: To improve understanding, we have provided a larger visualization and an expanded analysis in **Fig. 19 (Appendix A.7: Insights into SRAM)**.
> >   3. Clarification on the apparent artifact: The element in the second column of the first row of the 3x3 grid is not an artifact; it is the clothing logo. We have clarified this in the Appendix to avoid further confusion.
> > - We hope these updates improve the clarity of **Fig.1**.
>
> **Q2.  In the experiment described in Section 5.4, what loss functions were used to "densify attention distributions" and "improve semantic alignment"? A description of these would be helpful.**
> > - Following the comment, we have clarified the loss functions used in **Section 5.4**. From the SRAM corresponding to each generated patch of the initial custom model shown in **Fig.10**, we identified two main areas for improvement: *1) densifying attention distributions*, and *2) enhancing semantic alignment between the generated and reference patches*. To this end, we applied three loss functions in detail:
> >   1. **Distance-centering maximization loss** ($\\mathcal{L}\_{\\text{DCML}}$): Encourages the center coordinates of attention scores for generated patches to align with their corresponding reference patches, thus maximizing positional accuracy.
> >   2. **Total variation loss** ($\\mathcal{L}\_{\\text{TV}}$): Mitigates abrupt changes that may arise from $\\mathcal{L}\_{\\text{DCML}}$, ensuring smooth transitions and maintaining appropriate spacing between patches.
> >   3. **Center-weighted Gaussian loss** ($\\mathcal{L}\_{\\text{CWG}}$): Focuses dispersed attention scores around the central point of each patch, enhancing the focus of the attention map.
> > - Together, these loss functions improve patch correspondence and enhance the consistency and quality of generated images. Detailed descriptions of each loss function can now be found in **Appendix A.5 (Custom Model)**. We hope this updated explanation provides clarity regarding the objectives and implementation of these loss functions.

---

> > ### Comment · Reviewer_4XJm · 2024-11-24
> > **Thank you for your work and the detailed response.**
> >
> > After going through the authors' answers and reading the revised paper, I changed my initial assessment of the importance of this work. Specifically, in my earlier assessment I may have undervalued the contribution of this work providing a systematic approach for interpreting (cross-attention based) image-to-image diffusion models.
> >
> > Therefore, after careful re-consideration, I've decided  to increase my initial rating of *reject* to *marginally above the acceptance threshold*, while increasing the soundness of the work from *poor* to *good*, and the contribution from *poor* to *fair*.
> >
> > Besides, the revised version improves the presentation of the work, so I've decided to increase the presentation score from *fair* to *good*.
> >
> > Just as a minor point for further improvement I suggest the authors to denote the column-wise averaged attention scores differently from the attention scores defined in Equation (6).

---

> > > ### Author Response · Authors · 2024-11-25
> > > **To Reviewer 4XJm**
> > >
> > > We sincerely appreciate the tremendous effort you put into reviewing our paper. We are pleased to see that the significant concerns raised by the reviewer have been successfully addressed. Additionally, we will incorporate further improvements suggested during the discussion, such as clearly differentiating the column-wise averaged attention scores from the attention scores defined in **Equation (6) in the final version**. Thank you for your dedicated time and effort to carefully review our paper and provide constructive comments and positive feedback. Based on your insights, we have improved both the content and presentation of the paper, enhancing the accuracy and reliability of our research.

---

### Official Review · Reviewer_HMWz · 2024-11-03

**Soundness:** 3
**Presentation:** 3
**Contribution:** 2
**Rating:** 5
**Confidence:** 4

**Summary:**

This paper introduces a method called Image-to-Image Attribution Maps (I2AM) that enhances the interpretability of image-to-image (I2I) diffusion models by visualizing bidirectional attribution maps between the reference and generated images. I2AM aggregates cross-attention scores across time steps, attention heads, and layers, providing insights into feature transfers between images. It has been demonstrated to be effective in tasks like object detection, inpainting, and super-resolution, successfully identifying key regions for output generation even in complex scenes. Additionally, the paper introduces the Inpainting Mask Attention Consistency Score (IMACS), a new metric that assesses the alignment between attribution maps and inpainting masks, correlating strongly with existing performance metrics. Extensive experiments show that I2AM aids in model debugging and refinement, offering practical tools to improve the performance and interpretability of I2I models.

**Strengths:**

1. The paper introduces Image-to-Image Attribution Maps (I2AM), a novel method for enhancing the interpretability of image-to-image (I2I) diffusion models. I2AM visualizes bidirectional attribution maps, providing a clear view of feature transfers between reference and generated images.
2.Task Effectiveness: Demonstrates I2AM's effectiveness in object detection, inpainting, and super-resolution tasks.  Successfully identifies critical regions for output generation, even in complex scenes.
3.IMACS Metric: Introduces the Inpainting Mask Attention Consistency Score (IMACS), a new evaluation metric correlating with existing performance metrics.

**Weaknesses:**

1.The IMACS metric does not seem to be very relevant to the topic of this article, I2AM, which seems to be designed specifically for Inpainting and has no general use for other T2I tasks.
2.Some of the tasks selected in this article do not seem to be the most representative of I2I tasks, such as generative Inpating, image editing, image super-resolution, etc., so I have some concerns about the reliability and robustness of the conclusions in this article.

**Questions:**

1.The explanation in Figure 3 (b) is somewhat puzzling, why say "low-resolution signals result in lower involvement from early layers in controlling coarse features. " does not seem common sense, and why there is only one type of visualization in (a) and two in (b)?
2.Why did you choose PASD as the SR model? In fact, there are many influential generative SR models in the recent top conferences, like SUPIR, SeeSR. Will your approach come to the same conclusion on these models?

---

> ### Author Response · Authors · 2024-11-22
> **Response to Reviewer HMWz (1/3)**
>
> We sincerely appreciate the reviewer’s valuable feedback. Based on the provided comments, we have updated the experimental results and paper, with the changes highlighted in blue. If you have any further concerns, please feel free to let us know. We hope the responses are clear and kindly request that you consider a higher evaluation score.
>
> **W1. The IMACS metric does not seem to be very relevant to the topic of this article, $\\text{I}\^2\\text{AM}$, which seems to be designed specifically for Inpainting and has no general use for other T2I tasks.**
> > - We understand the concern regarding the relevance of IMACS to our method $\\text{I}\^2\\text{AM}$. To clarify, $\\text{I}\^2\\text{AM}$ is a general interpretability method applicable to a wide range of I2I tasks, while IMACS is specifically designed as a complementary evaluation metric tailored for inpainting tasks.
> > - $\\text{I}\^2\\text{AM}$ provides visual insights into the areas where the model focuses during I2I tasks, serving as a versatile attribution method across tasks such as object detection, super-resolution, and inpainting. In contrast, IMACS quantitatively evaluates attention alignment by using image masks to differentiate between attended and non-attended regions, which is particularly useful for inpainting tasks. Inpainting requires careful attention to masked regions and the integration of relevant features from the reference image, making alignment crucial for model performance. To achieve this, IMACS is designed to penalize attention to irrelevant areas and reward focus on the masked regions, effectively tracking this alignment. For example, as shown in **Tab. 4(below) in the Appendix**, increasing the penalty factor $\lambda$ for models with poor sore alignment (e.g., DCI-VTON) highlights IMACS’s ability to measure and address misaligned attention. This specificity makes IMACS less applicable to I2I tasks like object detection or super-resolution, where precise mask alignment is not as essential.
> > $$
> \\begin{array}{l|rr|rr|rr|rr}
> \\hline
>  & \\lambda=1 & & \\lambda=3 & & \\lambda=5 & (\text{default}) & \\lambda=10 &
> \\\\
> \text{Method} & \\text{IMACS}\_{\\textsf{g}}  \\uparrow &  \\text{IMACS}\_{\\textsf{r}}   \\uparrow & \\text{IMACS}\_{\\textsf{g}}   & \\text{IMACS}\_{\\textsf{r}}  & \\text{IMACS}\_{\\textsf{g}}  &  \\text{IMACS}\_{\\textsf{r}}  & \\text{IMACS}\_{\\textsf{g}}  &  \\text{IMACS}\_{\\textsf{r}}
> \\\\ \\hline
> \text{DCI-VTON} & 0.4291 & - & 0.0785 & - & -0.3051 & - & -1.2155 & - \\\\
> \text{StableVITON} & \\textbf{0.5376} & \\textbf{0.3534} & \\textbf{0.3083} & \\textbf{0.3388} & \\textbf{0.07} & \\textbf{0.3211} & \\textbf{-0.1219} & \\textbf{0.301} \\\\
> \text{Custom} & 0.2352 & 0.0677 & 0.0833 & 0.0403 & -0.1901 & 0.0091 & -0.4579 & -0.0527 \\\\
> \text{Refined custom} & \\underline{0.4675} & \\underline{0.1249} & \\underline{0.2215} & \\underline{0.0948} & \\underline{0.029}
> & \\underline{0.0513} & \\underline{-0.3105} & \\underline{0.0112} \\\\ \\hline
> \\end{array}
> $$
> > - Together, $\\text{I}\^2\\text{AM}$ and IMACS provide complementary benefits: $\\text{I}\^2\\text{AM}$ is broadly applicable across I2I tasks to enhance visual interpretability, while IMACS focuses on quantitatively assessing alignment in inpainting tasks. These approaches support a more comprehensive evaluation and understanding of model behavior.

---

> ### Author Response · Authors · 2024-11-22
> **Response to Reviewer HMWz (2/3)**
>
> **W2. Some of the tasks selected in this article do not seem to be the most representative of I2I tasks, such as generative Inpainting, image editing, image super-resolution, etc., so I have some concerns about the reliability and robustness of the conclusions in this article.**
> > - We understand your concerns regarding the reliability and robustness of the study, as well as the representativeness of the selected tasks. While this research focuses on generative inpainting, object detection, and super-resolution tasks, we believe that $\\text{I}\^2\\text{AM}$ can be extended to other I2I tasks, such as colorization, depth estimation, style transfer, and image translation, especially when these tasks utilize cross-attention mechanisms to process reference images.
> > - We acknowledge that the use of cross-attention mechanisms in I2I tasks is currently limited, and this has been now discussed as a limitation in the **Conclusion of the main text**. As mentioned in the **Introduction**, many existing latent diffusion models (LDMs) [1, 7, 8, 9, 13] process reference images by simply concatenating or adding them to input images. This approach has drawbacks when it comes to isolating and visualizing the individual contributions of the reference and input images from an explainability perspective. In contrast, the cross-attention mechanism that is of interest in this work independently learns the relationships between the reference and input images, emphasizing their interactions, which significantly enhances interpretability.
> > - Recently, methods that utilize cross-attention for reference image processing have garnered attention. These include the introduction of new image encoders [2, 6, 10, 11, 12, 14] and the addition of cross-attention layers to ControlNet [3, 4, 5]. These approaches address the limitations of concatenation or addition operations above, allowing more effective visualization of the interactions between reference images and generated elements. Although these methods are not yet mainstream, they show great potential for advancing I2I tasks and underscore the applicability of $\\text{I}\^2\\text{AM}$ to a wider range of tasks.
> > - Thus, while this study has focused on specific tasks, we are confident that the conclusions drawn are reliable and robust. We remain optimistic about the future applicability of $\\text{I}\^2\\text{AM}$ across a broader set of tasks.
>
> [1] Zhang, Lvmin, Anyi Rao, and Maneesh Agrawala. "Adding conditional control to text-to-image diffusion models." *Proceedings of the IEEE/CVF International Conference on Computer Vision*. 2023.
>
> [2] Yang, Binxin, et al. "Paint by example: Exemplar-based image editing with diffusion models." *Proceedings of the IEEE/CVF Conference on Computer Vision and Pattern Recognition*. 2023.
>
> [3] Kim, Jeongho, et al. "Stableviton: Learning semantic correspondence with latent diffusion model for virtual try-on." *Proceedings of the IEEE/CVF Conference on Computer Vision and Pattern Recognition*. 2024.
>
> [4] Yang, Tao, et al. "Pixel-aware stable diffusion for realistic image super-resolution and personalized stylization." *arXiv preprint arXiv:2308.14469* (2023).
>
> [5] Wu, Rongyuan, et al. "Seesr: Towards semantics-aware real-world image super-resolution." *Proceedings of the IEEE/CVF conference on computer vision and pattern recognition*. 2024.
>
> [6]Kim, Kangyeol, et al. "Reference-based image composition with sketch via structure-aware diffusion model." *arXiv preprint arXiv:2304.09748* (2023).
>
> [7] Saharia, Chitwan, et al. "Image super-resolution via iterative refinement." *IEEE transactions on pattern analysis and machine intelligence* 45.4 (2022): 4713-4726.
>
> [8] Meng, Chenlin, et al. "Sdedit: Guided image synthesis and editing with stochastic differential equations." *arXiv preprint arXiv:2108.01073* (2021).
>
> [9] Choi, Jooyoung, et al. "Ilvr: Conditioning method for denoising diffusion probabilistic models." *arXiv preprint arXiv:2108.02938* (2021).
>
> [10] Wang, Qixun, et al. "Instantid: Zero-shot identity-preserving generation in seconds." *arXiv preprint arXiv:2401.07519* (2024).
>
> [11] Ye, Hu, et al. "Ip-adapter: Text compatible image prompt adapter for text-to-image diffusion models." *arXiv preprint arXiv:2308.06721* (2023).
>
> [12] Wang, Chenhui, et al. "FLDM-VTON: Faithful Latent Diffusion Model for Virtual Try-on." *arXiv preprint arXiv:2404.14162* (2024).
>
> [13] Li, Leheng, et al. "OmniBooth: Learning Latent Control for Image Synthesis with Multi-modal Instruction." *arXiv preprint arXiv:2410.04932* (2024).
>
> [14] Seyfioglu, Mehmet Saygin, et al. "Diffuse to Choose: Enriching Image
> Conditioned Inpainting in Latent Diffusion Models for Virtual Try-All." *arXiv preprint arXiv:2401.13795* (2024).

---

> ### Author Response · Authors · 2024-11-22
> **Response to Reviewer HMWz (3/3)**
>
> **Q1. The explanation in Figure 3 (b) is somewhat puzzling, why say "low-resolution signals result in lower involvement from early layers in controlling coarse features. " does not seem common sense, and why there is only one type of visualization in (a) and two in (b)?**
> > - First, **Fig. 3** has been corrected to **Fig. 2** to reflect the proper mentioned order. We appreciate your point regarding the explanation in **Fig. 2(b)** and have made revisions to improve clarity. In general, latent diffusion models (LDMs) perform inference starting from pure noise, resulting in clear representations and feature attention scores in the early layers, as shown in **Fig. 2(a)**. However, some super-resolution (SR) models [1, 2] combine low-quality (LQ) images with pure noise for better denoising.
> > - The PASD [1] used in our experiments follows this approach. Visualization results indicate that the introduction of low-frequency signals during denoising leads to the phenomenon where "*low-resolution signals result in lower involvement from early layers in controlling coarse features*".  To clarify, we revised the description as follows (**L201-L203**):
> > > “*since LQ (low-quality) input signals are provided with the initial noise, the early layers seem to interpret the input as already containing basic information, leading them to pay less attention to coarse features. Further details are provided in the Appendix A.7.*”
> > - Additionally, to demonstrate that this phenomenon is not unique to PASD, we include the result of LLAM for SeeSR [2], another SR model, in **Fig. 18**.
> > - Finally, **Fig. 2(a) and (b)** show the same type of visualization results (specifically, **Layer-Level Attribution Maps**). However, since PASD uses more layers ($9$ vs. $15$) compared to other models, we offer more layer visualizations.
>
> [1] Yang, Tao, et al. "Pixel-aware stable diffusion for realistic image super-resolution and personalized stylization." *arXiv preprint arXiv:2308.14469* (2023).
>
> [2] Wu, Rongyuan, et al. "Seesr: Towards semantics-aware real-world image super-resolution." *Proceedings of the IEEE/CVF conference on computer vision and pattern recognition*. 2024.
>
> **Q2. Why did you choose PASD as the SR model? In fact, there are many influential generative SR models in the recent top conferences, like SUPIR, SeeSR. Will your approach come to the same conclusion on these models?**
> > - $\\text{I}\^2\\text{AM}$ facilitates the visualization of models that use reference images in cross-attention modules, which is why PASD was chosen. Thank you for mentioning SUPIR and SeeSR. We reviewed these models and found that SUPIR does not incorporate reference images through cross-attention mechanisms, making it unsuitable for $\\text{I}\^2\\text{AM}$ visualization. On the other hand, SeeSR meets this criterion, allowing its generation process to be visualized using $\\text{I}\^2\\text{AM}$. A detailed comparison of PASD and SeeSR is now provided in **Appendix (A.7: Comparison of PASD and SeeSR)**.
> > - As explained in response to **Q1**, both SeeSR and PASD insert low-quality (LQ) latents into pure noise during the denoising process. Previous works [1,2] indicate that this method mitigates the train-test discrepancy caused by the Signal-to-Noise Ratio (SNR) not reaching zero at the final time step in Stable Diffusion's noise schedule. Our experiments revealed similar trends in both PASD and SeeSR, with some differences arising due to variations in the amount of low-frequency information during the inference process, as explained below. Specifically, SeeSR, which utilizes stronger LQ input signals, focuses on narrower reference regions compared to PASD (refer to **Figures 8, 9 vs. Figures 17, 18**). The initial noise configurations for each model are as follows:
> >   - Inference in general LDMs: $\\mathbf{z}\_{T} = \\mathbf{\\epsilon} \\sim \\mathcal{N}(0,I)$
> >   - $\\mathbf{z}\_{T}'$ and $\\mathbf{z}\_{T}''$ are independent Gaussian noises $\\epsilon \\sim \\mathcal{N}(0,I)$; $\\mathbf{z}\_{\\text{LR}}$ is LQ latent.
> >   - SeeSR: $\\mathbf{z}\_T^{\\text{seesr}} = \\sqrt{\\bar \\alpha\_{T}} \\mathbf{z}\_{\\text{LR}} + \\sqrt{1 - \\bar \\alpha\_{T}} \\mathbf{z}\_{T}'$
> >   - PASD: $\\mathbf{z}\_{T}\^{\\text{pasd}} = \\sqrt{\\bar \\alpha\_{a} \\bar \\alpha\_{T}} \\mathbf{z}\_{\\text{LR}} + \\sqrt{1 - \\bar\\alpha\_{a} \\bar \\alpha\_{T}} \\mathbf{z}\_{T}''$; introduce $\\bar\\alpha\_{a} = 0.1189$ to control the residual signal $\\mathbf{z}\_{\\text{LR}}$
>
> [1] Choi, Jooyoung, et al. "Perception prioritized training of diffusion models." *Proceedings of the IEEE/CVF Conference on Computer Vision and Pattern Recognition*. 2022.
>
> [2] Lin, Shanchuan, et al. "Common diffusion noise schedules and sample steps are flawed." *Proceedings of the IEEE/CVF winter conference on applications of computer vision*. 2024.

---

> ### Author Response · Authors · 2024-12-02
> **Official Comment by Reviewer HMWz**
>
> **Dear Reviewer HMWz**,
>
> As the ICLR discussion phase is nearing its conclusion, we would like to gently remind you to review our responses to the comments and questions you raised. Your feedback is highly valued and will be crucial in refining our work. We would greatly appreciate any additional thoughts or suggestions you may have.
>
> Thank you again for your time and valuable insights throughout the review process. We look forward to your response.
>
> Best regards,
> *Authors of Submission 5933*

---

### Public Comment · ~Zhiwei_Jia1 · 2024-11-17
**A useful way to interpret diffusion models for image-to-image tasks**

Hi authors,

Thanks for the work, which I think provides a useful way to interpret diffusion models for image-to-image tasks. Since the paper focuses on the paired image-to-image translation task, I wonder if it can be extended to the unpaired translation tasks where certain properties unique to the unpaired setup make content consistency a harder issue, as discussed in [1].

[1] Semantically Robust Unpaired Image Translation for Data With Unmatched Semantics Statistics, ICCV 2021

---

> ### Author Response · Authors · 2024-11-22
> **Reply to Comments from Zhiwei Jia**
>
> **Q. Since the paper focuses on the paired image-to-image translation task, I wonder if it can be extended to the unpaired translation tasks where certain properties unique to the unpaired setup make content consistency a harder issue, as discussed in [1].**
> > - Thank you for the thoughtful feedback and for recognizing the value of our work. In response to your comments, we have extended our method to address unpaired image-to-image translation tasks, where maintaining content consistency can be more challenging due to the potential for semantic flips, as discussed in [1].
> > - In this rebuttal, we included an analysis of the unpaired setting in **Fig.21**. In unpaired translation tasks, semantic flips may occur to ensure distribution consistency, which means the model could invert the semantics of the input [1]. To evaluate our method in this context, we applied $\\text{I}\^2\\text{AM}$
>  to visualize attribution maps under unpaired conditions. As shown in **Fig.21**, the attribution maps display similar trends to those observed in the paired setting. This indicates that our method remains effective even in unpaired setups, providing reliable interpretability. Additionally, we have addressed this discussion and its limitations in the **Conclusion**, also highlighting future opportunities to extend our method to broader scenarios.
>
> [1] Semantically Robust Unpaired Image Translation for Data With Unmatched Semantics Statistics, ICCV 2021

---

### Author Response · Authors · 2024-12-04
**To all Reviewers**

We sincerely thank all reviewers for their time and effort in evaluating our paper.

In this study, we propose a novel methodology to enhance the interpretability of image-to-image diffusion models. Our approach introduces a bidirectional attribution map that utilizes the spatial structure of the reference image to reveal relationships between generated images. Responding to requests for comparisons with other attention-based methods, we provide a detailed analysis highlighting the differences between our approach and attention maps commonly used in text-to-image tasks (e.g., DAAM [1]).

To address the reviewers’ questions and incorporate their feedback, we summarize the revisions and additional experiments conducted during the rebuttal period as follows:

**Additional Experiments**
> 1. Additional experiments with the super-resolution model SeeSR *(Reviewer HMWz Q2, N4o3 W2)*
>    - Comparative analysis with the existing SR model, PASD.
>    - Visualizations using $\\text{I}\^2\\text{AM}$ show consistent results despite variations due to differences in inference mechanisms.
> 2. Comparison experiments with the T2I-based attribution method (DAAM) *(Reviewer 4XJm W1, N4o3 W1, bmkU W2)*
>    - Differences such as the direction of Softmax are highlighted in Tab. 1 and Fig. 16.
> 3. Expanded version of Fig. 1 *(Reviewer 4XJm Q1)*
>    - Addressing concerns about readability due to the small size of the original figure.
>    - Larger visualizations and additional analysis are provided for clarity.
> 4. Ablation study on $\\lambda$ and $\\delta$ *(Reviewer N4o3 W4, bmkU W1)*
> 5. Addition of limitations and related experiments *(Reviewer 4XJm W2, bmkU W3)*
>    - Experiments analyzing two axes simultaneously (e.g., head-layer, time-layer).
>    - Validation of robustness through $\\text{I}\^2\\text{AM}$ visualizations in unpaired settings.

**Clarifications**
> 1. Potential of $\\text{I}\^2\\text{AM}$ with increased use of cross-attention mechanisms in I2I *(Reviewer HMWz W2)*
> 2. Methodological differences between T2I and I2I through comparisons with DAAM *(Reviewer 4XJm W1, N4o3 W1, bmkU W2)*
> 3. Additional explanations for each figure *(Reviewer HMWz Q1, 4XJm W3, W5)*
>    - Enhancements to improve readability and understanding.
> 4. Corrections to IMACS equations *(Reviewer 4XJm W4)*
> 5. Detailed descriptions of unified/layer/head-level maps and SRAM, along with expected benefits *(Reviewer 4XJm W2, zNnp W1, N4o3 W3)*
> 6. Conditions for model visualization *(Reviewer zNnp W2)*
> 7. Specification of the loss function used in Section 5.4 *(Reviewer 4XJm Q2)*

These revisions and enhancements aim to improve the clarity and completeness of our paper. Once again, we deeply appreciate the valuable feedback provided by the reviewers.

[1] Tang, Raphael, et al. "What the daam: Interpreting stable diffusion using cross attention." arXiv preprint arXiv:2210.04885 (2022).

---

### Meta-Review · Area_Chair_T5of · 2024-12-19

**Metareview:**

This work studied the cross attention maps in I2I models and proposed a new method for interpreting LDMs which enables debugging and enhances the performance in various tasks. All reviewers appreciate the novelty of studying I2I models unlike prior arts in T2I models and the value of findings in other tasks. It receives three marginally above and two marginally below accept. Meanwhile, several concerns such as clarification of novelty, missing baselines, insufficient visualizations, missing ablation study are raised by reviewers. Author did a great job by providing detailed pinpoint feedback and two reviewers upgrade the score after clarifying some misunderstandings. AC agrees that the findings with I2I model is of great value as few works target on that and more importantly, these findings are demonstrated to be useful to improve performance in downstream tasks. This would benefit a lot to the field of diffusion models. After considering all the comments and discussions, a decision of acceptance is made and authors are advised to revise and incorporate all valuable comments from reviewers in final draft to make the work more complete.

**Additional Comments On Reviewer Discussion:**

Several concerns such as clarification of novelty, missing baselines, insufficient visualizations, missing ablation study are raised by reviewers. Authors provide detailed pinpoint feedback and two reviewers upgrade the score after clarifying some misunderstandings. While three reviewers are positive, the other two reviewers who gave the score of 5 (marginally below) did not provide feedback for author's rebuttal. AC carefully checked the rebuttal and thinks most concerns (no critical one found) are well addressed. Thus, a final decision of acceptance is made.

---

### Decision · Program_Chairs · 2025-01-22

Accept (Poster)